# AutoFold: Protein Generation via Autoregressive Contact Graph Generation

## Abstract

Generative models for protein design, particularly diffusion and flow matching approaches, are powerful but computationally expensive, with slow sampling times that hinder high-throughput applications. We introduce AutoFold, an ultra-fast autoregressive model that generates proteins via a sparse graph representation of their structure. Instead of generating continuous coordinates directly, AutoFold learns a contact graph of the backbone structure. A Vector Quantized Variational Autoencoder is trained to discretize contacting inter-residue geometric features, creating a graph representation with single edge labels invariant to SE(3) transformations. This representation can be decoded to reconstruct a backbone structure with high fidelity. We then train an autoregressive model to generate these graphs, further incorporating amino acid sequence into node attributes. Our trained model can be seamlessly used for both unconditional generation and motif scaffolding. Our results demonstrate that AutoFold achieves unique co-designability comparable to state-of-the-art methods while accelerating sampling by 2-3 times and generating highly diverse and novel proteins. By shifting generation from continuous coordinates to discrete graphs, AutoFold introduces a novel modeling paradigm for protein design.

## 1 Introduction

The *de novo* design of proteins holds transformational promise to solve long-standing challenges in biomedicine and material science (Huang et al., 2016; Kuhlman & Bradley, 2019; Korendovych & DeGrado, 2020). A central problem in this field is modeling the joint distribution of a protein's amino acid sequence and its three-dimensional (3D) structure. While recent approaches have made significant progress by generating sequence and structure separately (Watson et al., 2023; Ingraham et al., 2023; Dauparas et al., 2022), co-generating them is crucial for precisely controlling the arrangement and interaction of structural elements. This joint modeling problem is inherently difficult, as it requires navigating a mixed discrete-continuous space of sequences and structures.

To this end, recent methods, particularly diffusion (Campbell et al., 2024; Chu et al., 2024) and flow matching models (Geffner et al., 2025a), have advanced the field by learning to jointly refine a random cloud of atoms and a random or fully masked sequence, represented as either a continuous or a discrete variable, into a valid protein. Despite their power, these state-of-the-art models share a critical limitation: computational cost. Their iterative refinement process, which requires hundreds or thousands of denoising steps, leads to slow sampling times. Additionally, these models often rely on computationally expensive equivariant modules (Jumper et al., 2021), further slowing down their sampling process (Watson et al., 2023). This computational bottleneck severely hinders their application in high-throughput settings. The resulting trade-off between design quality and scale of generation might significantly constrain the pace and scope of protein design discovery.

In this work, we introduce AutoFold, an ultra-fast, autoregressive generative model that circumvents these limitations by reformulating the design problem. Instead of operating directly on continuous 3D coordinates, AutoFold learns to generate a sparse, discrete graph representation of protein structure from which both sequence and 3D coordinates can be efficiently recovered. Our approach consists of two stages. First, we train a Vector Quantized Variational Autoencoder (VQ-VAE) to learn a finite "vocabulary" of codes that describe the geometric relationship between contacting residue pairs. This model transforms a continuous protein backbone into a sparse attributed graph where

nodes represent residues and edges are labeled with a discrete code capturing the SE(3)-invariant geometry of the contact. This graph can be decoded back to a full 3D backbone structure with high fidelity (0.8Å RMSD). In the second stage, we train a powerful autoregressive model on this graph representation, additionally treating the sequence as node attributes, to generate novel protein graphs. Decoding the generated graph yields both its sequence and structure.

By shifting from iterative refinement in a continuous space to autoregressive generation in a discrete space, AutoFold dramatically accelerates the design process. We demonstrate its effectiveness on two fundamental design tasks: unconditional sequence-structure co-generation and motif scaffolding, where a functional motif is extended with a surrounding protein structure. Our results show that AutoFold not only achieves sampling speed over an order of magnitude faster than state-of-the-art diffusion models but also produces high-quality, highly diverse and novel designs with performance comparable to diffusion or flow matching models. Moreover, we identify a significant pitfall in the current evaluation pipeline, where extreme helix-rich samples can inflate novelty while not being detected by the standard co-designability or diversity metrics. By breaking the speed and novelty barrier, AutoFold opens the door to large-scale computational protein design, accelerating the discovery of proteins with highly novel structures.

Our primary contributions can be summarized as follows:

- **A novel protein graph representation with high fidelity:** We introduce a sparse, SE(3)-invariant graph representation for protein backbones learned via a VQ-VAE, which simplifies the geometry of protein structures into a discrete and computationally tractable format.
- **Ultra-fast autoregressive generation:** We develop an autoregressive model, termed Auto-Fold, that generates these protein graphs, enabling fast sampling.
- **Competitive performance:** We demonstrate that AutoFold achieves performance comparable to state-of-the-art diffusion and flow matching models for both unconditional generation and motif scaffolding tasks.
- **Open-source implementation and datasets:** We will release our models, code, and the novel datasets of protein graphs to facilitate future research in the community.

## 2 RELATED WORK

Our work is situated at the intersection of two rapidly evolving fields: protein structure generation and graph generative modeling. In this section, we review key developments in both areas to contextualize our contribution.

**Protein Structure Generation.** While generative models for protein design have achieved remarkable success, progress has been dominated by computationally intensive diffusion and flow matching methods. Seminal works like Chroma (Ingraham et al., 2023), RFDiffusion (Watson et al., 2023) established the power of these approaches, inspiring a cascade of models, such as FrameDiff (Yim et al., 2023), FoldFlow (Bose et al., 2024), and others (Trippe et al., 2023; Wu et al., 2024), for protein backbone generation. A significant leap in performance came from scaling up training data using the AlphaFold database (AFDB), as demonstrated by models like FoldFlow2 (Huguet et al., 2024), Genie (Lin & Alquraishi, 2023), Genie2 (Lin et al., 2024), and Proteína (Geffner et al., 2025b). The field has since then progressed towards the co-design of sequence and structure and even fully atomistic generation (Campbell et al., 2024; Chu et al., 2024; Ren et al., 2024; Lisanza et al., 2023; Fu et al., 2024; Yim et al., 2025; Geffner et al., 2025a), with language models like ESM3 (Hayes et al., 2025) and DPLM2 (Wang et al., 2025) also emerging as powerful alternatives. Despite this rapid progress, reliance on computationally expensive paradigms has persisted, except for a recent work (Jendrusch & Korbel, 2025) focusing on efficient generation but still based on iterative refinement. Our work departs from this trend, introducing, to our knowledge, the first autoregressive model for *de novo* protein sequence and structure co-generation. *By shifting the generative task from continuous 3D coordinates to a discrete, sparse graph representation*, we directly address the computational bottlenecks that have limited prior methods.

**Graph Generative Models.** Early deep learning approaches for graph generation were predominantly autoregressive, sequentially constructing graphs by adding nodes and edges, as pioneered by GraphRNN (You et al., 2018) and GRAN (Liao et al., 2019). While foundational, these meth-

ods were often surpassed in performance by recent diffusion models like DiGress (Vignac et al., 2023), which operate via a global noising-denoising process on the graphs. Our work builds upon AutoGraph (Chen et al., 2025a), a modern autoregressive framework that addresses the architectural limitations of its predecessors. AutoGraph bridges the scalability of sequence modeling with the complexity of graph generation by "flattening" graphs into sequences via random walks, enabling a decoder-only transformer to learn the distribution. We adapt this highly efficient framework to the specific domain of protein contact graphs, leveraging its efficiency for rapid structure generation.

## 3 BACKGROUND

In this section, we first provide an overview of rigid-body transformations, or "frames", which are fundamental for representing the geometry of protein structures. We then revisit the principles of autoregressive generative models for graphs, which form the basis of our generation process.

### 3.1 FRAMES IN PROTEIN STRUCTURES

To describe the precise 3D arrangement of a protein structure, we represent the local coordinate system of each residue as a frame. A frame is a mathematical object that encapsulates both the position (translation) and orientation (rotation) of a rigid body in 3D space. We use a formulation similar to that of Ingraham et al. (2019); Hayes et al. (2025), defining the frame $T_i \in \mathrm{SE}(3)$ for residue $i$ as a $4 \times 4$ transformation matrix:

$$\mathbf{T}_i = \begin{bmatrix} \mathbf{R}_i & \mathbf{t}_i \\ 0_{1 \times 3} & 1 \end{bmatrix} \in \mathrm{SE}(3),$$

where $\mathbf{R}_i \in \mathrm{SO}(3)$ is a rotation matrix and $\mathbf{t}_i$ is a transition vector.

The translation vector, $\mathbf{t}_i$, simply specifies the global position of the residue's $\alpha$-carbon ($C_\alpha$) atom. The rotation matrix, $\mathbf{R}_i$, defines the residue's orientation. It is constructed from an orthonormal basis derived from the backbone atoms $(N, C_\alpha, C)$, which effectively aligns the residue's local coordinate system with a global reference. This allows for the representation of all local atomic positions in a standardized way, independent of the protein's overall orientation.

Using these frames, we can transform a point $p_{\mathrm{local}}$ from the local coordinate system of residue $i$ to the global coordinate system, and vice versa:

- Local to global transformation: $p_{\mathrm{global}} = \mathbf{R}_i p_{\mathrm{local}} + \mathbf{t}_i$
- Global to local transformation: $p_{\mathrm{local}} = \mathbf{R}_i^\top (p_{\mathrm{global}} - \mathbf{t}_i)$

This formalism is essential for creating geometric representations that are invariant to global rotations and translations, a critical property for learning meaningful structural patterns.

### 3.2 AUTOREGRESSIVE GRAPH GENERATIVE MODELS

Autoregressive models are a class of generative models that produce complex data structures, such as graphs, by sequentially generating their components. They factorize the joint probability distribution over the entire structure into a product of conditional probabilities. The AutoGraph framework (Chen et al., 2025a) adapts this principle for scalable graph generation by first "flattening" a graph into a sequence representation. This transformation allows the complex problem of graph generation to be reframed as a sequential language modeling task, making it amenable to powerful and highly scalable architectures like the Transformer (Vaswani et al., 2017).

Its core principle is to define a stochastic ordering of nodes and edges through some random walk sampling with restarts and neighborhood information, creating a sequence $s = (s_1, s_2, \ldots, s_n)$ where each token $s_i$ represents either a node index, a node or edge label, or a special token. The model then learns the probability of the graph $G = (V, E)$ as a product of conditional probabilities over this sequence:

$$p(G) = \prod_{i=1}^{n} p(s_i \mid s_{<i}).$$

In this formulation, the generation process unfolds one step at a time, predicting the next token (which models the act of visiting a neighbor, restarting at a new node, or indicating whether the

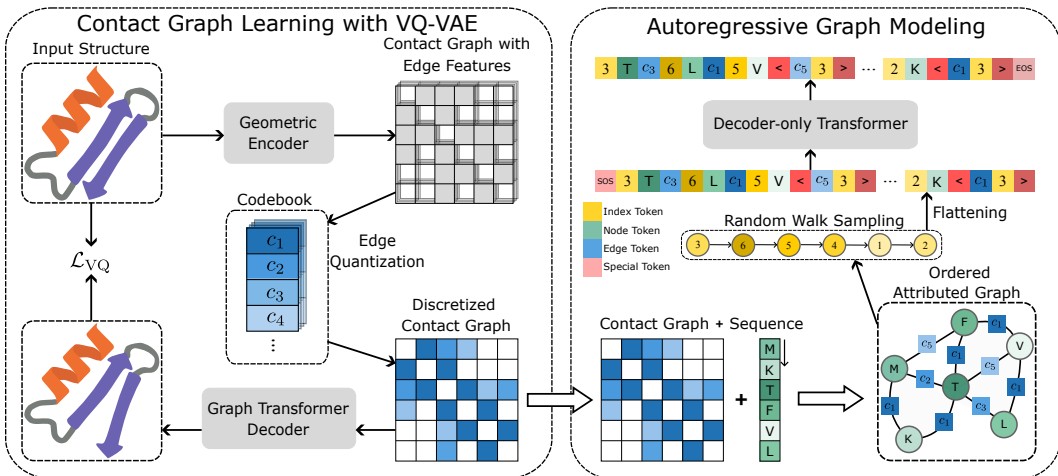

Figure 1: AutoFold is a two-stage generative model for protein design. Stage I (left): It leverages VQ-VAE to learn a sparse contact graph with single-edge labels for each protein structure that can be decoded to reconstruct the 3D backbone coordinates with high fidelity; Stage II (right): It uses the AutoGraph framework (Chen et al., 2025a) to generate these discrete graphs autoregressively by "flattening" them into sequential representations via random walk sampling. The random walks are allowed to have restarts and are coupled with a neighborhood set for each node. The flattening of the random walk introduces special tokens, such as '/' to indicate a restart between segments, and '<' and '>' to mark the start and end of a neighborhood set. More details are given in Appendix C.4.

current node is a neighbor of a previously visited node) based on the previously generated sequence. By leveraging a decoder-only transformer to model these sequential dependencies, this approach can capture the long-range relationships inherent in graph structures, making it well-suited for modeling the complex topology of protein contact graphs.

## 4 THE AUTOFOLD MODEL

Our generative model for protein design is a two-stage process that first learns a discrete graph representation of a protein's structure using a VQ-VAE, and then generates these graphs using an autoregressive model. The entire pipeline of our model is illustrated in Figure 1.

### 4.1 PROTEIN CONTACT GRAPH LEARNING WITH VQ-VAE

We represent a protein structure as a sparse contact graph $G = (V, E)$, where nodes $v \in V$ correspond to residues, and an edge $e \in E$ exists between any pair of residues whose $C_\alpha$ atoms are within a distance threshold of 8.0 Å, a standard choice for the contact in the literature (Rao et al., 2021; Lin et al., 2023). Each edge is then encoded into a discrete structural token, creating a graph with single-labeled edges that can be decoded to reconstruct the 3D backbone coordinates. While previous work has demonstrated that protein structures can be compressed into token sequences (Hayes et al., 2025), our method is the first to show they can be effectively compressed into contact graphs that *preserve local geometry* and reconstruct the original structure with high fidelity. This local preservation is critical for downstream tasks such as motif scaffolding. This compression is learned using a VQ-VAE, which consists of an encoder, a discrete codebook, and a decoder.

#### 4.1.1 ENCODER

The VQ-VAE encoder operates independently on each pair of contacting residues $(i, j) \in E$ to produce a continuous embedding vector $z_{ij} \in \mathbb{R}^d$ that is *invariant to SE(3) transformations* on the backbone coordinates. This embedding is derived from the residues' local frames and their relative sequence positions using geometrically invariant operators. Relative sequence positions between contacting residues are first clamped to a maximum of $\pm 32$, meaning long-range contacts share the same positional embeddings. These positions are then embedded into vectors, defining the initial edge states $x_{ij} \in \mathbb{R}^d$. This position encoding is commonly adopted in prior work (Jumper et al.,

2021). Note that the encoder input is purely structural: no amino acid sequence information is used at this stage. These initial states are then processed through a series of geometric encoder blocks.

Inspired by Hayes et al. (2025), each geometric layer extracts invariant features that describe the relative orientation and position of the two residues by learning anchor points within their local frames. Specifically, for an edge $x_{ij}$ and its corresponding endpoint frames $\mathbf{T}_i$ and $\mathbf{T}_j$, each geometric layer computes its output through four steps (pseudocode is provided in Appendix C.1):

1. **Anchor Projections:** Two sets of anchor points, $(r_i, r_j)$ and $(d_i, d_j)$, are linearly projected from the input $x_{ij}$, where the projections' parameters are learnable. All have a shape of $\mathbb{R}^{h \times 3}$, where $h$ is the number of anchor heads, a hyperparameter analogous to attention heads.
2. **Global Frame Conversion:** The anchor points, initially in their local residue frames, are transformed into the global rotational and distance frames: $r_i = \mathbf{R}_i(r_i), r_j = \mathbf{R}_j(r_j) \in \mathbb{R}^{h \times 3}$ and $d_i = \mathbf{T}_i(d_i), d_j = \mathbf{T}_j(d_j) \in \mathbb{R}^{h \times 3}$.
3. **Distance and Direction between Anchors:** The rotational similarity $R$ for each head is calculated as the dot product $R_{ij,h} = 1/\sqrt{3}r_{i,h} \cdot r_{j,h}$. The distance for each head is the $L_2$ norm $D_{ij,h} = \|d_{i,h} - d_{j,h}\|_2$. Similarly to Ingraham et al. (2019), each distance $D_{ij,h}$ is then lifted using a radial basis function (16 RBF kernels isotropically spaced from 0 to 12 Å) and projected back to its original dimension.
4. **Output Projection:** The final output $\text{GeomLayer}(x_{ij}, \mathbf{T})$ is a linear projection of the concatenated vector $[R_{ij}; D_{ij}] \in \mathbb{R}^{2h}$ into $\mathbb{R}^d$.

A geometric encoder block, similar to the transformer block in ESM3 (Hayes et al., 2025), uses Pre-LayerNorm and a SwiGLU activation:

$$x = x + \text{GeomLayer}(x, \mathbf{T}) \in \mathbb{R}^{|E| \times d}, \quad x = x + \text{SwiGLUMLP}(x) \in \mathbb{R}^{|E| \times d}.$$

**Symmetrization of Edges.** Without symmetrization, the above geometric encoder blocks would create a bidirected graph with different edge features for each direction of the edge. To make the generative task easier, we find that symmetrizing the graph before the last linear projection layer is quite useful: $z_{ij} = z_{ij} + z_{ji}$. And later in the decoder, we could desymmetrize the graph through a linear transformation of the lower-triangular edges: $z_{ij} = z_{ij}$ if $i < j$ and $z_{ij} = \text{Linear}(z_{ij})$ if $i > j$. We show in Appendix G.1 that this approach does not affect the reconstruction quality.

### 4.1.2 CODEBOOK LEARNING VIA VECTOR QUANTIZATION

The continuous embedding $z_{ij} \in \mathbb{R}^d$ from the encoder (after some downprojection) is mapped to the nearest vector in a learned, finite codebook $C = \{c_k\}_{k=1}^K \in \mathbb{R}^{K \times d}$, where $K$ is the number of discrete token types. This quantization step yields a discrete token index $k_{ij}$ for each edge:

$$k_{ij} := \arg\min_k \|z_{ij} - c_k\|_2.$$

The set of indices $\{k_{ij}\}_{(i,j) \in E}$ for all contacting pairs constitutes the final discrete graph representation. A commitment loss with a coefficient of 0.25 is employed. To ensure stable training and effective codebook utilization, we update the codebook using an exponential moving average of the encoder outputs and re-initialize unused codes (Van Den Oord et al., 2017; Razavi et al., 2019; Roy et al., 2018; Hayes et al., 2025).

### 4.1.3 DECODER

The decoder reconstructs the 3D backbone coordinates from the discrete graph representation. It takes the quantized codebook vectors $\{c_{k_{ij}}\}_{(i,j) \in E}$ as input and processes them through a stack of message passing blocks followed by a stack of transformer blocks, an architecture inspired by Graph Transformers (Wu et al., 2021; Chen et al., 2022). The resulting residue representations are passed to a projection head that predicts the final 3D coordinates for the $N$, $C_\alpha$, and $C$ atoms.

The entire VQ-VAE is trained end-to-end by minimizing a composite loss function $\mathcal{L}_{\text{VQ}}$ following ESM3 (Hayes et al., 2025). In addition to the VQ commitment loss in Section 4.1.2, this includes the other five losses. The primary losses are geometric distance and geometric direction, which supervise the reconstruction of high-quality backbone structures. Binned distance and direction classification losses serve as auxiliary objectives to bootstrap training. Finally, an inverse folding

token prediction loss is used to encourage the learned representations to be informative for sequence-based tasks. Further details on the loss functions are provided in Appendix C.3. Note that all-atom position decoding is also possible by using a two-stage training process by Hayes et al. (2025).

## 4.2 AUTOREGRESSIVE GRAPH GENERATION

Once the structure is transformed into a graph with single-labeled edges, the second stage of our model learns to generate these graphs autoregressively. This stage further incorporates the amino acid sequence into the graph as node labels. By simultaneously generating both edge and node labels, our model enables co-design of both the structure and sequence.

Following the AutoGraph framework (Chen et al., 2025a), we generate graphs by first "flattening" them into a sequential representation and then modeling this sequence autoregressively. We generate a stochastic ordering of nodes and their labeled edges through a random path with restarts and neighborhood information along the path. The neighborhood information is represented as a set of all previously visited nodes that are neighbors of any node in the path. This process creates a token sequence $s = (s_1, \ldots, s_n)$, where each token $s_i$ can represent a node index, a node or edge label, or some special token. The special tokens include '/' to indicate a restart between path segments, and '<' and '>' to mark the start and end of a neighborhood set. The node index token captures the graph topology, while the node and edge labels represent the amino acids and the geometry of the contacting residues. We provide full details of this process in Appendix C.4. A key advantage of AutoGraph over other autoregressive models is its ability to perform *substructure-conditioned generation*, making it feasible for motif scaffolding without any specific tuning. We note that the stochastic ordering in the flattening process is used only to define an autoregressive factorization over graphs, not to impose invariance. Additionally, as a protein has a *canonical ordering* given by its sequence, we use this ordering as the node indexing in this sequence. We employ a decoder-only transformer architecture, namely the Llama model (Touvron et al., 2023a;b), to model these sequences autoregressively, allowing it to learn the complex dependencies inherent within proteins.

## 4.3 INFERENCE AND MOTIF SCAFFOLDING

During inference, we generate novel protein structures by sampling from the trained transformer model. The process begins with a starting token (SOS), and the model autoregressively predicts the next token in the sequence until an end-of-sequence token (EOS) is generated. We employ standard sampling strategies, such as top-$p$ (nucleus) sampling with annealed temperature, to control the diversity and quality of the generated sequences. In particular, since our model has different token types, we use different temperatures when sampling different types of tokens. More details can be found in Appendix D.4. Additionally, we perform constrained decoding (Scholak et al., 2021) to ensure that the generated sequence is syntactically valid, allowing it to be convertible into a meaningful protein graph. Once a complete sequence is sampled, it is deterministically converted back into a contact graph and subsequently into 3D backbone coordinates using the trained decoder described in Section 4.1.3.

This approach supports two primary generation modalities: (1) Unconditional co-generation of sequence and structure: The process is initiated with an empty context, allowing the model to generate a complete graph from scratch, representing a novel protein structure and sequence. (2) Motif scaffolding: To build a structure around a predefined motif, the fixed motif is first converted into its corresponding graph representation through the trained VQ-VAE encoder. Note that, as our VQ-VAE encoder is *locally preserved*, its graph representation will be the same as its subgraph representation when it is within any protein structures. This subgraph is then "flattened" into a seed sequence using the same random walk procedure as in training. This seed sequence is provided as a prompt to the autoregressive model, which then generates the remainder of the sequence, effectively building the scaffold structure around the fixed motif.

## 5 EXPERIMENTS

We evaluate AutoFold on unconditional co-generation of sequence and structure, and motif scaffolding tasks. Since our model has a two-stage process, we first assess the reconstruction quality of the discrete graph representation learned by the VQ-VAE.

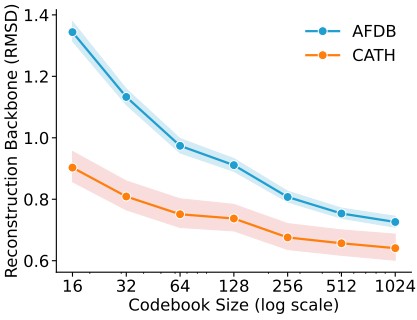 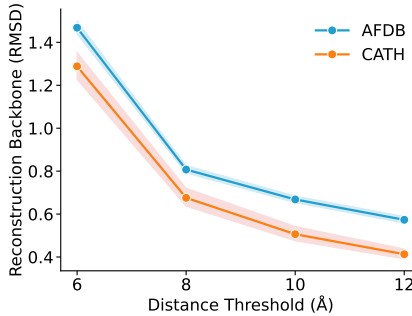

Figure 2: Backbone reconstruction RMSD for VQ-VAE models. The RMSDs are computed on two different datasets: a held-out AFDB validation set and the CATH test set of experimental structures. Left: reconstruction errors vs. the codebook size in VQ-VAE. Right: reconstruction errors vs. the distance threshold used to construct the graphs. 95% confidence intervals are shown by shading.

**Experimental Setup.** Following prior work (Lin et al., 2024; Geffner et al., 2025b;a), we train all our models on the dataset of Foldseek AFDB clusters (Van Kempen et al., 2024; Barrio-Hernandez et al., 2023), containing 588,318 protein structures predicted by AlphaFold2 (Jumper et al., 2021). These proteins are sufficiently diverse, as represented by cluster representatives from the structure-based Foldseek. Like prior work, we only keep protein lengths below 256 residues in our experiments. We randomly split the data into train and validation sets with a ratio of 90% and 10%. More dataset details are provided in Appendix D.1.

To evaluate the reconstruction quality of the VQ-VAE models, we also use the CATH test dataset (Orengo et al., 1997; Ingraham et al., 2019) in addition to the AFDB validation set, which contains 1120 experimental structures. This dataset enables testing the generalizability of our VQ-VAE model, which is trained entirely on structures predicted by AlphaFold2.

## 5.1 AUTOFOLD LEARNS HIGH-FIDELITY GRAPH REPRESENTATION OF STRUCTURE

We first validate the reconstruction quality of our VQ-VAE model. The goal of VQ-VAE is to compress the continuous protein backbones into a sparse, discrete graph representation with minimal loss of information. We measure the fidelity using the backbone atom root mean square deviation (RMSD) between the original and the reconstructed structures. Our evaluation focuses on two key hyperparameters: the codebook size, which determines the richness of the geometric vocabulary, and the contact distance threshold, which controls the graph's sparsity.

First, to assess the expressive power of our discrete vocabulary, we fix the contact threshold at 8.0 Å and vary the codebook size. As shown on the right of Figure 2, reconstruction RMSD improves monotonically as the codebook size increases. A modest codebook of 16 codes yields a reasonable RMSD of under 1.5 Å, while a large codebook of 1024 achieves high-fidelity, sub-angstrom reconstructions with a mean RMSD of 0.7 Å on AFDB. Note that this reconstruction metric should not be directly compared to La-Proteina's, which reconstructs only non-$C_\alpha$ coordinates given the $C_\alpha$ coordinates and the latent variables, a much easier task. Instead, this result is comparable to ESM3's VQ-VAE (Figure S3.B (Hayes et al., 2025)), which also learns discrete backbone latents.

Next, we investigate the impact of graph sparsity by fixing the codebook size to 256 and varying the distance threshold. As expected, creating denser graphs with a larger threshold further reduces reconstruction error. However, this also increases the complexity for the subsequent autoregressive generation task. As our goal is to find a usefully sparse contact graph representation that can be used for generation, rather than maximizing the reconstruction quality, we select a distance threshold of 8.0 Å, a common choice for creating *sparse* contact graphs (Lin et al., 2023), resulting in an average degree of around 8. We fix the codebook size at 256 to maintain a good balance between reconstruction quality and generation difficulty, as increasing the size makes the generative task harder. For example, the same motif might be encoded into a graph with completely different edge labels due to an overused number of codes.

## 5.2 Unconditional Sequence and Structure Co-generation

Since AutoFold is the only autoregressive model in the field, we benchmark it against state-of-the-art diffusion and flow matching models for joint sequence and structure generation. The baseline methods include P(all-atom) (Qu et al., 2024), APM (Chen et al., 2025b), PLAID (Lu et al., 2024), ProteinGen (Lisanza et al., 2024), Protpardelle (Chu et al., 2024), DPLM-2 (650M) (Wang et al., 2025), MultiFlow (Campbell et al., 2024), and La-Proteina (Geffner et al., 2025a). For each baseline method, we generate 100 proteins for each length in $\{100, 200, 300, 400, 500\}$ following Geffner et al. (2025a). We train three AutoFold models with different sizes, namely AutoFold-t (27M), AutoFold-s (116M), and AutoFold-m (407M), following the typical model size settings of Llama (Touvron et al., 2023b). More details about their hyperparameters are provided in Appendix D.4. Since autoregressive models are length unconstrained, we randomly generate 500 proteins for AutoFold models without explicit length controls. While this is not equivalent to evaluating on generated sets of proteins of fixed lengths, it provides a middle ground to compare AutoFold, the first autoregressive model, to diffusion and flow matching models. We additionally provide length distributions of generated proteins in Appendix E.4. We acknowledge that generating long co-designable proteins, such as La-Proteina, remains challenging, and we leave this for future research. We assess performance using a comprehensive set of established metrics, including co-designability, diversity, novelty (against PDB and AFDB), and standard designability. Co-designability evaluates how well co-generated sequences fold into generated structures, while designability uses ProteinMPNN (Dauparas et al., 2022) to produce sequences for generated structures divided by the total number of generated samples. Additionally, we include two new metrics: (1) unique co-designability, which represents the number of clusters given by Foldseek (Van Kempen et al., 2024) for the co-designable protein structures; (2) sequence FID, which measures the Fréchet distance between the AFDB and generated sequences, embedded with ESM2 (Lin et al., 2023). As observed by Geffner et al. (2025a), the current dataset leads to models that generate proteins with a low $\beta$-sheet content. Therefore, we additionally train AutoFold on the secondary-structure-filtered subset used by La-Proteina (denoted AFDB-LaProteina), yielding AutoFold-t⋆, -s⋆, -m⋆. For inference hyperparameters, including top-$p$ sampling and temperatures, we select them either to optimize co-designability or increase $\beta$-sheet content. We note that users can readily tune them to match their specific needs in practice. More details about evaluation can be found in Appendix D.3.

**Performance on Design Quality.** Our results in Table 1 demonstrate that when tuned to maximize co-designability, AutoFold-m achieves better unique co-designability and much better diversity and novelty compared to the baselines. Additionally, AutoFold achieves better sequence quality compared to La-Proteina, with a sequence FID of 10.18, 10.09, and 9.07 for t, s, and m models, respectively, versus 10.31 for La-Proteina. We note that our ProteinMPNN-8 designability is lower than La-Proteina's, and sometimes lower than AutoFold's co-designability. This is expected as there is a trade-off between designability and novelty. In fact, highly novel backbones are, by construction, more challenging for a sequence-design model such as ProteinMPNN, which is trained on native protein structures. This difficulty can then propagate to ESMFold, leading to lower designability even if the generated backbones are physically plausible.

These models exhibit a strong preference for $\alpha$-helical structures, with helical content reaching approximately 90%. Our analysis suggests that this helix overrepresentation primarily stems from two sources: inference hyperparameters and the training dataset. Our results quantify and partially mitigate both effects. First, when we retune decoding to favor $\beta$-sheets on the original AFDB dataset, AutoFold-m reaches $\beta$-sheet proportions comparable to prior work ($\approx 3\text{–}4\%$, see Appendix C.1 in Geffner et al. (2025a)) while maintaining unique co-designability. Second, AutoFold-m⋆, trained on AFDB-LaProteina, attains a secondary structure profile similar to La-Proteina-2, indicating that the dataset is the dominant driver. We provide a few examples of co-designable proteins generated by AutoFold-m⋆ in Figure 4 and additional examples generated by all models in Appendix F.1.

Lastly, we remark that extreme helix-rich samples can inflate novelty by producing long helices: AutoFold models tuned to optimize co-designability tend to generate many long $\alpha$-helices (Appendix F.1), which are detected by FoldSeek as highly diverse and novel. We attract the community's attention to this critical evaluation issue and encourage its further elucidation. To more fairly compare against previous models, one should compare AutoFold-m⋆ to La-Proteina-2, both with a similar helices/sheets proportion. Our model's diversity and novelty remain clearly better than La-Proteina. We hypothesize that this could be attributed to the discrete graph representation we use,

Table 1: Comparison of methods across co-designability, normalized diversity, novelty, designability, and secondary structure proportions. AutoFold-m* was trained on AFDB-LaProteina, the same secondary-structure filtered AFDB subset used by La-Proteina. La-Proteina-1/2 correspond to the same model using two different inference hyperparameters, where La-Proteina-2 achieves a higher beta-sheet content. These metrics are highly coupled and should be interpreted jointly.

| Method | Co-designability (%) ↑ | | Diversity (normalized) ↑ | | | Novelty ↓ | | Designability (%) ↑ | Sec. Struct. (%) | |
|---|---|---|---|---|---|---|---|---|---|---|
| | Unique | all | Str | Seq | Seq+Str | PDB | AFDB | PMPNN-8 | $\alpha$ | $\beta$ |
| P(all-atom) | 26.8 | 37.9 | 0.707 | 0.781 | 0.871 | 0.72 | 0.81 | 57.9 | 56 | 17 |
| APM | 6.4 | 32.2 | 0.199 | 0.398 | 0.366 | 0.84 | 0.89 | 61.8 | 73 | 8 |
| PLAID | 5.0 | 19.2 | 0.260 | 0.312 | 0.396 | 0.89 | 0.92 | 37.6 | 44 | 14 |
| ProteinGen. | 2.4 | 17.8 | 0.135 | 0.315 | 0.270 | 0.83 | 0.89 | 54.2 | 78 | 5 |
| Protpardelle | 2.0 | 35.2 | 0.057 | 0.210 | 0.119 | 0.79 | 0.82 | 56.2 | 65 | 14 |
| DPLM-2 (650M) | 14.2 | 29.2 | 0.486 | 0.541 | 0.520 | 0.95 | 0.96 | 53.2 | 40 | 19 |
| MultiFlow | 37.8 | 71.2 | 0.531 | 0.685 | 0.846 | 0.83 | 0.88 | 93.6 | 72 | 12 |
| La-Proteina-1 | 41.2 | **72.2** | 0.571 | 0.598 | 0.834 | 0.75 | 0.82 | 93.8 | 72 | 5 |
| La-Proteina-2 | 36.0 | 59.6 | 0.604 | 0.634 | 0.836 | 0.77 | 0.86 | **94.6** | 63 | 10 |
| *Tuned to optimize co-designability* | | | | | | | | | | |
| AutoFold-t | 36.2 | 50.4 | **0.718** | 0.984 | 0.952 | 0.16 | 0.19 | 58.0 | 89 | 2 |
| AutoFold-s | 42.4 | 68.8 | 0.616 | 0.910 | 0.945 | **0.10** | **0.12** | 69.4 | 91 | 2 |
| AutoFold-m | **46.4** | 72.0 | 0.644 | 0.950 | **0.986** | 0.14 | 0.17 | 69.6 | 90 | 1 |
| *Tuned to maximize Beta-sheet content* | | | | | | | | | | |
| AutoFold-t | 29.2 | 45.4 | 0.643 | **0.987** | 0.956 | 0.16 | 0.19 | 52.2 | 89 | 2 |
| AutoFold-s | 37.6 | 55.0 | 0.618 | 0.942 | 0.978 | 0.19 | 0.23 | 57.0 | 88 | 2 |
| AutoFold-m | 39.2 | 58.2 | 0.674 | 0.952 | 0.966 | 0.24 | 0.26 | 60.6 | 85 | 4 |
| AutoFold-t* | 15.0 | 23.0 | 0.652 | 0.861 | 0.887 | 0.44 | 0.51 | 33.4 | 75 | 11 |
| AutoFold-s* | 24.4 | 39.4 | 0.619 | 0.853 | 0.939 | 0.56 | 0.60 | 46.4 | 70 | 13 |
| AutoFold-m* | 32.0 | 48.0 | 0.667 | 0.950 | 0.967 | 0.48 | 0.52 | 46.8 | 69 | 12 |

which allows AutoFold to move into novel structure space while staying within a learned manifold of locally valid backbones, leading to high novelty without catastrophic loss of foldability.

**Generation Speed.** A primary advantage of AutoFold is its generation speed. To provide a direct comparison with length-conditioned models, we timed the generation process using constrained decoding, where generation is halted once a target length is reached. As shown in Figure 3, both AutoFold-s and AutoFold-m are substantially faster than the state-of-the-art baselines. Most notably, AutoFold accelerates generation by more than two orders of magnitude compared to RFDiffusion, a widely used and representative diffusion model.

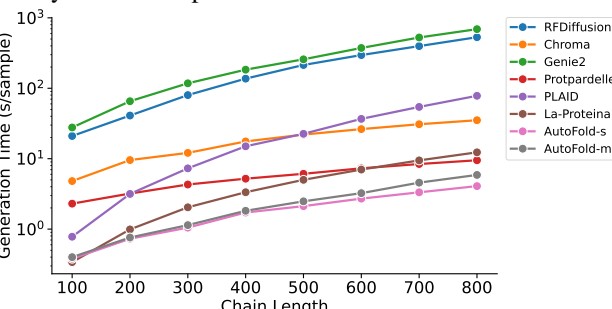

Figure 3: The runtimes for different models for maximum batch size obtained on a single NVIDIA H100 80GB GPU.

Figure 4: Structures generated by AutoFold-m*. Blue: generated, gray: ESMFold predicted.

## 5.3 MOTIF SCAFFOLDING

A key advantage of our autoregressive model is its intrinsic ability to perform motif scaffolding *without re-training or fine-tuning*. Because our model generates protein graphs sequentially, we can provide a functional motif as a fixed "prompt" and simply let the model autoregressively complete the surrounding scaffold. This zero-shot capability is a direct benefit of our modeling strategy.

**Experimental Setup.** To assess this capability, we evaluate AutoFold on the motif scaffolding benchmark from Geffner et al. (2025a). Following their protocol, a design is considered successful

Table 2: Motif scaffolding benchmark: number of successful designs for AutoFold and baselines

| Motif | # Segments | Protpardelle | | La-Proteina | | AutoFold-m | | AutoFold-long | |
|---|---|---|---|---|---|---|---|---|---|
| | | All | Unique | All | Unique | All | Unique | All | Unique |
| 1YCR | 1 | 1 | 1 | 123 | 38 | 68 | 44 | 89 | **52** |
| 3IXT | 1 | 0 | 0 | 34 | **6** | 11 | **6** | 8 | 4 |
| 4ZYP | 1 | 0 | 0 | 11 | 2 | 23 | **8** | 9 | 5 |
| 5TPN | 1 | 0 | 0 | 55 | 1 | 1 | 1 | 3 | **2** |
| 5WN9 | 1 | 0 | 0 | 0 | 0 | 2 | **1** | 0 | 0 |
| 5TRV_short | 1 | 0 | 0 | 5 | 1 | 4 | **3** | 3 | 2 |
| 5TRV_med | 1 | 0 | 0 | 65 | **3** | 0 | 0 | 1 | 1 |
| 5TRV_long | 1 | 0 | 0 | 91 | **9** | 0 | 0 | 0 | 0 |
| 6E6R_short | 1 | 0 | 0 | 35 | 8 | 23 | **16** | 7 | 4 |
| 6E6R_med | 1 | 0 | 0 | 73 | **22** | 8 | 4 | 1 | 1 |
| 6E6R_long | 1 | 0 | 0 | 71 | **43** | 1 | 1 | 4 | 1 |
| 7MRX_60 | 1 | 0 | 0 | 7 | 3 | 20 | **5** | 1 | 1 |
| 7MRX_85 | 1 | 0 | 0 | 16 | **4** | 8 | 1 | 2 | 1 |
| 7MRX_128 | 1 | 0 | 0 | 22 | **17** | 2 | 1 | 1 | 1 |
| 1PRW | 2 | 0 | 0 | 175 | **20** | 162 | 16 | 176 | 11 |
| 2KL8 | 2 | 80 | **1** | 165 | **1** | 200 | **1** | 196 | **1** |
| 4JHW | 2 | 0 | 0 | 2 | 1 | 1 | 1 | 17 | **4** |
| 5IUS | 2 | 0 | 0 | 16 | 2 | 52 | 9 | 67 | **15** |
| 6VW1 | 2 | 0 | 0 | 21 | 1 | 95 | 6 | 126 | **12** |
| 1BCF | 4 | 70 | 1 | 189 | 7 | 157 | 9 | 190 | **12** |
| Total | | 151 | 3 | 1176 | **189** | 838 | 133 | 901 | 130 |
| Total (w/o fixed length) | | 151 | 3 | 791 | 79 | 772 | 102 | 881 | **118** |

if it is co-designable and the generated scaffold aligns to the native motif with a backbone RMSD < 1 Å. A current limitation of our method lies in scaffolding sparse active sites. If the distances between key functional residues in a motif exceed our 8.0 Å contact distance cutoff, their geometric relationship cannot be encoded into the graph representation. This prevents the model from correctly scaffolding the motif. For this reason, we excluded active site motifs from our evaluation, resulting in 20 tasks. We believe this limitation could be addressed by increasing the contact distance threshold to capture longer-range interactions, a direction we leave for future research. To demonstrate the benefit of a larger distance threshold, we include AutoFold-long, a variant that uses a 9.0 Å cutoff with only a negligible increase in sampling time. More results are provided in Appendix F.2, G.2.

**Results.** As summarized in Table 2, AutoFold-m shows highly competitive performance, solving 18 out of 20 motif scaffolding tasks. It performs on par with or better than the state-of-the-art model, La-Proteina, on 13 tasks in terms of producing unique valid designs. As a length-unconstrained model, AutoFold is not designed for tasks with rigid length constraints, and thus it fails on the two fixed-length challenges in the benchmark (5TRV_long and 5TRV_med). However, since precise length control is often unnecessary for practical design applications, this is not a significant drawback. In fact, when evaluated on flexible-length tasks (all benchmarks excluding 5TRV, 6E6R, and 7MRX), AutoFold-m outperforms La-Proteina by generating a greater number of unique successful designs (102 vs 79). Finally, the AutoFold-long variant further improves upon these results, yielding even more unique successful designs (118 vs 102), particularly for two-segment motifs.

## 6 CONCLUSION

We have introduced AutoFold, an ultra-fast autoregressive model that solves the critical speed bottleneck in *de novo* protein design. By reformulating the problem from continuous coordinate refinement to discrete graph generation, AutoFold achieves a fundamental shift in methodology. Our model demonstrates performance competitive with state-of-the-art models in generating high-quality, diverse proteins and enables zero-shot motif scaffolding without any task-specific fine-tuning, all while being over an order of magnitude faster than leading diffusion and flow matching approaches. As a by-product, our work also produces a rich set of protein graphs that can be readily used to boost the development of graph generative models.

## ETHICS STATEMENT

The research conducted in this paper is purely computational and relies on protein structure data from the publicly available Protein Data Bank and AlphaFold database, which is intended for scientific use. Our work aims to accelerate scientific discovery for beneficial applications, such as the design of novel therapeutics, enzymes, and materials. We acknowledge that, like any powerful generative technology, protein design models could theoretically be misused for harmful purposes. However, our research is intended solely for positive advancements in science and medicine. By developing more efficient computational tools, we hope to reduce the energy footprint associated with large-scale protein design campaigns.

## REPRODUCIBILITY STATEMENT

All code for AutoFold models, along with the final trained model weights, will be made publicly available on GitHub upon publication. The datasets of protein graphs generated and used in this study will also be released. The pseudocode and architecture details are provided in Appendix A. The underlying protein structure data is sourced from the cluster representatives of the FoldSeek (Van Kempen et al., 2024) clustered version of the AlphaFold database, and all processing details are provided in Appendix D.1 to ensure that our dataset construction can be fully reproduced.

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

# Appendix

## CONTENTS

## A    LIMITATIONS

While AutoFold demonstrates a powerful new paradigm for ultra-fast protein co-design, several limitations present clear avenues for future improvement.

**Architectural and Representational Choices**    Our architectural design is centered on a sparse contact graph with a fixed small $C_\alpha$ distance threshold. While this choice balances reconstruction fidelity with generative tractability, it introduces several limitations. This choice restricts the model's ability to capture long-range interactions, which is critical for scaffolding sparse active sites. It also contributes to an observed structural bias towards $\alpha$-helices, as their local contact patterns are more easily represented by our VQ-VAE. Future work should explore alternative graph representations, such as using larger distance thresholds, k-nearest neighbors, or different graph generative models to capture a wider range of structural motifs and long-range dependencies.

**Dataset Considerations**    Our training dataset, originally used in Lin et al. (2024), was not curated with advanced filters for secondary structure balance, unlike methods such as La-Proteina (Geffner et al., 2025a). Implementing such data curation strategies could directly mitigate the $\alpha$-helical bias and improve the diversity of generated secondary structures.

**Beyond Unconditional and Motif Scaffolding**    While we have demonstrated success in unconditional generation and motif scaffolding, AutoFold's framework could be extended to a broader range of design challenges. Future applications could include designing large protein complexes and protein binders, and incorporate explicit structural constraints to achieve target folds or functional properties.

## B    USE OF LARGE LANGUAGE MODELS

This work used large language models in the following ways:

**Preparation of plots**    LLMs were partly used to generate the code for the plots presented in this work. The correctness of all code and data was checked manually. The data shown in the figures was generated by manually written code.

**Code development**    LLMs were used to assist in developing evaluation pipelines using the Cursor IDE. All code modified by LLMs was manually checked.

**Polishing of manuscript**    LLMs were occasionally used to refine or rephrase individual sentences.

## C    ARCHITECTURE AND TRAINING DETAILS

We provide additional details of our model architectures, loss functions, and optimization details.

### C.1    VQ-VAE ENCODER

We provide the full pseudocode for the geometric layer in Algorithm 2, which uses the RBF function presented in Algorithm 1. The output of the encoder is then down-projected onto a smaller dimensional space before performing the vector quantization.

---

**Algorithm 1** $\text{RBF}(D)$

---

**Input:** Distances $D \in \mathbb{R}^h$, $n_{\text{bin}} \in \mathbb{N}_+$, $D_{\min}, D_{\max} \in \mathbb{R}$
**Output:** $D_{\text{rbf}} \in \mathbb{R}^{h \times n_{\text{bin}}}$
1: $D_\mu = \text{linspace}(D_{\min}, D_{\max}, n_{\text{bin}})$                $\triangleright \mathbb{R}^{n_{\text{bin}}}$
2: $\sigma = D_{\max} - D_{\min}/n_{\text{bin}}$
3: $D_{\text{rbf},i} = \exp(-\|D_i - D_\mu\|^2/\sigma^2)$             $\triangleright \mathbb{R}^{h \times n_{\text{bin}}}$

---

---

**Algorithm 2** GeomLayer($x$, **T**)

---

**Input:** Contact graph $G = (V, E)$, input edge features $x \in \mathbb{R}^{|E| \times d}$, frames $\mathbf{T} \in \mathrm{SE}(3)^{|V|}$
**Output:** Output edge features $x \in \mathbb{R}^{|E| \times d}$

1: Anchor projections:
$\quad\quad r_i, r_j, d_i, d_j = \mathrm{Linear}(x_{ij})$ for any $(i, j) \in E$ $\qquad\qquad\qquad \triangleright (\mathbb{R}^{h \times 3})_{\times |E| \times 4}$
2: $(\mathbf{R}_i, \mathbf{t}_i) = \mathbf{T}_i$ for any $i \in V$ $\qquad\qquad\qquad\qquad\qquad\qquad \triangleright (\mathrm{SO}(3)^{|V|}, \mathbb{R}^{|V| \times 3})$
3: Global rotational frame:
$\quad\quad r_i = \mathbf{R}_i(r_i), r_j = \mathbf{R}_j(r_j)$ for any $(i, j) \in E$ $\qquad\qquad\qquad \triangleright (\mathbb{R}^{h \times 3})_{\times |E| \times 2}$
4: Global distance frame:
$\quad\quad d_i = \mathbf{T}_i(r_i), d_j = \mathbf{T}_j(r_j)$ for any $(i, j) \in E$ $\qquad\qquad\qquad \triangleright (\mathbb{R}^{h \times 3})_{\times |E| \times 2}$
5: Direction between anchors:
$\quad\quad R_{ij,h} = {^1}/{\sqrt{3}} r_{i,h} \cdot r_{j,h}$ for any $(i, j) \in E$ $\qquad\qquad\qquad\qquad \triangleright \mathbb{R}^{|E| \times h}$
6: Distance between anchors:
$\quad\quad D_{ij,h} = \mathrm{Linear}(\mathrm{RBF}(\|d_{i,h} - d_{j,h}\|_2))$ for any $(i, j) \in E$ $\qquad \triangleright \mathbb{R}^{|E| \times h}$
7: Output projection:
$\quad\quad x_{ij} = \mathrm{Linear}([R_{ij}; D_{ij}])$ for any $(i, j) \in E$ $\qquad\qquad\qquad\qquad \triangleright \mathbb{R}^{|E| \times d}$

---

## C.2 VQ-VAE DECODER

The decoder consists of a stack of message passing blocks followed by a stack of bidirectional transformer blocks inspired by recent graph transformer architectures (Wu et al., 2021; Chen et al., 2022). The node features are initialized with zeroes, and are then fed into the decoder jointly with the dequantized edge features.

**Message Passing Blocks.** The message passing blocks use an attention mechanism to update the node and edge representations after each block. Specifically, for input node and edge features $h \in \mathbb{R}^{|V| \times d}$ and $x_{ij} \in \mathbb{R}^{|E| \times d}$, each message passing block first computes $\mathrm{MessagePassing}(h, x, G)$ by performing the following attention-based message passing:

$$h_Q, h_K, h_V = \mathrm{Linear}(h) \in \mathbb{R}^{|V| \times d_{\mathrm{hidden}}}$$

$$x_{QK} = \mathrm{Linear}(x) \in \mathbb{R}^{|E|} \text{ and } x_V = \mathrm{Linear}(x) \in \mathbb{R}^{|E| \times d_{\mathrm{hidden}}}$$

$$\alpha_{ij} = \frac{\exp w_{ij}}{\sum_{k \in \mathcal{N}(i)} w_{ik}} \text{ with } w_{ij} = \frac{h_{Q,i} \cdot h_{K,j} + x_{QK,ij}}{\sqrt{d_{\mathrm{hidden}}}} \text{ for any } (i, j) \in E$$

$$h_i = \sum_{j \in \mathcal{N}(i)} \alpha_{ij}(h_{V,j} + x_{V,j}) \in \mathbb{R}^{d_{\mathrm{hidden}}}$$

$$h = \mathrm{Linear}(h) \in \mathbb{R}^{|V| \times d}.$$

Similar to the original self-attention mechanism (Vaswani et al., 2017), we also use the multi-head attention. Additionally, it has an edge updater $\mathrm{EdgeUpdater}(x, h, G)$:

$$x'_{ij} = \mathrm{Linear}(h_i) + \mathrm{Linear}(h_j) + \mathrm{Linear}(x_{ij})$$

$$x_{ij} = x_{ij} + \mathrm{SwiGLUMLP}(x'_{ij}) \text{ for any } (i, j) \in E.$$

An entire message passing block performs the following operations:

$$h = h + \mathrm{MessagePassing}(h, x, G),$$

$$h = h + \mathrm{SwiGLUMLP}(h),$$

$$x = \mathrm{EdgeUpdater}(x, h, G).$$

**Transformer Blocks.** After the message passing blocks, a stack of bidirectional transformer blocks with RoPE (Su et al., 2024) is employed. Each block uses the same architecture as in the ESM3 (Hayes et al., 2025) model, taking the node representations $h \in \mathbb{R}^{|V| \times d}$ as input:

$$h = h + \mathrm{MultiHeadAttention}(h),$$

$$h = h + \mathrm{SwiGLUMLP}(h).$$

**Prediction Head.** The final output of the decoder follows ESM3's `gram_schmidt` procedure (Algorithm 8 in Hayes et al. (2025)) to produce the reconstructed backbone structure coordinates.

## C.3 VQ-VAE Loss Function

The entire VQ-VAE is trained end-to-end by minimizing a composite loss function following ESM3 (Hayes et al., 2025):

$$\mathcal{L}_{\text{VQ}} = \mathcal{L}_{\text{dist}} + \mathcal{L}_{\text{dir}} + \mathcal{L}_{\text{binned dist}} + \mathcal{L}_{\text{binned dir}} + \mathcal{L}_{\text{inverse folding}} + 0.25 \mathcal{L}_{\text{commit}}, \tag{1}$$

where $\mathcal{L}_{\text{dist}}$ and $\mathcal{L}_{\text{dir}}$ are geometric distance and direction losses, which are primary losses for supervising the reconstruction of backbone structures. Additionally, binned distance and direction classification losses ($\mathcal{L}_{\text{binned dist}}$ and $\mathcal{L}_{\text{binned dir}}$) are used to bootstrap structure prediction. To produce the pairwise logits, we follow Algorithm 9 in ESM3 (Hayes et al., 2025). Moreover, an inverse folding token prediction loss ($\mathcal{L}_{\text{inverse folding}}$) is used to encourage the learned representations to contain information pertinent to sequence-related tasks. Finally, a VQ commitment loss ($\mathcal{L}_{\text{commit}}$) is used to encourage the learning of the codebook. We provide below the details of each loss function.

**Backbone Distance Loss $\mathcal{L}_{\text{dist}}$.** $\mathcal{L}_{\text{dist}}$ first computes the pairwise $L_2$ distance matrix for the predicted and true coordinates of the three backbone atoms $(N, C_\alpha, C)$. Let $D_{\text{pred}}, D \in \mathbb{R}^{3|V| \times 3|V|}$ be the corresponding distance matrices. We compute

$$\mathcal{L}_{\text{dist}} = \text{mean}(\min((D_{\text{pred}} - D)^2, 25))$$

**Backbone Direction Loss $\mathcal{L}_{\text{dir}}$.** It first computes six vectors for both predicted and ground truth coordinates for each residue: (a) $N \to C_\alpha$; (b) $C_\alpha \to C$; (c) $C \to N_{\text{next}}$; (d) $-(N \to C_\alpha) \times (C_\alpha \to C)$; (e) $(C_{\text{prev}} \to N) \times (N \to C_\alpha)$; (f) $(C_\alpha \to C) \times (C \to N_{\text{next}})$. Then, it computes the pairwise dot product between these vectors for both predicted and ground truth coordinates, denoted as $D_{\text{pred}}, D \in \mathbb{R}^{6|V| \times 6|V|}$. Finally, we compute

$$\mathcal{L}_{\text{dir}} = \text{mean}(\min((D_{\text{pred}} - D)^2, 20))$$

**Binned Distance Classification Loss $\mathcal{L}_{\text{binned dist}}$.** This loss bins the true distances between residues (specifically, their $C_\beta$) to get ground truth targets and computes a cross-entropy loss between these targets and pairwise logits. Specifically, we first calculate the location of $C_\beta$ based on $(N, C_\alpha, C)$ for the ground truth coordinates. Then, we compute the pairwise distance between $C_\beta$ and bin them into one of 64 bins, with lower bounds $[0, 2.3125^2, (2.3125 + 0.3075)^2, \ldots, 21.6875^2]$, forming labels $y \in \{0, \ldots, 63\}^{|V| \times |V|}$. Finally, we compute the pairwise logits using the last layer representations and Algorithm 9 in ESM3 and compute the cross-entropy loss using the labels $y$ and the logits.

**Binned Direction Classification Loss $\mathcal{L}_{\text{binned dir}}$.** Similar to the above loss, this loss captures a coarser similarity between ground truth and predicted orientations to stabilize early training. Specifically, we compute the pairwise dot product between three vectors $C_\alpha \to C$, $C_\alpha \to N$, and $(C_\alpha \to C) \times (C_\alpha \to N)$ normalized to unit length. Then, we bin these dot products into 16 evenly spaced bins in $[-1, 1]$, forming classification labels $y \in \{0, \ldots, 15\}^{|V| \times |V|}$. Finally, we compute the pairwise logits as above and compute the cross-entropy loss using the labels $y$ and the logits.

**Inverse Folding Loss $\mathcal{L}_{\text{inverse folding}}$.** We pass the final layer representations of the decoder through a regression head to produce logits. Then, we use ground truth amino acids as labels $y$, compute cross-entropy for the classification task of predicting amino acids from the final layer representations.

**VQ Commitment Loss $\mathcal{L}_{\text{commit}}$.** This is the commonly used loss for the VQ learning. Following the notation in Section 4.1.2, we compute

$$\mathcal{L}_{\text{commit}} = \frac{1}{|E|} \sum_{ij} \|z_{ij} - c_{k_{ij}}\|_2^2$$

## C.4 Autoregressive Graph Generation

We employ the AutoGraph framework (Chen et al., 2025a) to generate the protein contact graphs autoregressively. For each graph $G = (V, E)$, AutoGraph samples from this graph a random path sequence with restarts and neighborhood information $w = (w_1, \ldots, w_n)$, termed Segmented Eulerian Neighborhood Trail (SENT), where $w_k = (v_k, A_k)$ with a node (index) $v_k \in V$ and $A_k \subseteq V$ is the set of all previously visited nodes that are neighbors of $v_k$. This path (with restarts) visits each node and edge exactly once, making it a concise and lossless sequential representation of the graph. More importantly, each prefix of this sequence generates an *induced subgraph* of $G$, making it possible for substructure-conditioned generation. To model this sequence using a language model, we first convert it into a machine-readable sequence of tokens using the technique from Section 2.4 of AutoGraph (Chen et al., 2025a) with special tokens. These tokens include symbols such as '/' to indicate a breakage between segments, and '$<$' and '$>$' to mark the start and end of a neighborhood set. The resulting tokenization induces a non-Markovian random walk in the graph, incorporating additional virtual nodes labeled with the above special tokens. Language modeling of SENTs aims to learn the state transition probabilities of the random walks. Standard cross-entropy loss was used for training with teacher-forcing.

At inference time, we perform constrained decoding (Scholak et al., 2021) to ensure that the generated sequence is syntactically valid, allowing it to be convertible into a meaningful protein graph. For downstream tasks with length constraints, such as motif scaffolding, we restrict the allowable index tokens to a prescribed range (*e.g.,* in $\{1, ..., L_{\max}\}$). This guarantees that the model cannot generate indices beyond $L_{\max}$, and thus cannot produce proteins longer than the benchmark length cap. Conversely, minimum-length requirements can be enforced by adding complementary constraints (e.g., requiring that all indices up to $L_{\min}$ be generated before EOS). Because these constraints operate directly on the index-token stream, they integrate cleanly with our semantic validity constraints for graph decoding.

## C.5 Optimization Details

All our models are trained using the AdamW optimizer (Loshchilov & Hutter, 2019) with a standard linear warm-up strategy. The cosine learning rate decay scheduler is used for all models. The optimization hyperparameters are provided in Section D.4.

# D Experimental Details

## D.1 Datasets

We use one dataset to train all our models, including both VQ-VAE and autoregressive models. This dataset is based on the cluster representatives of the Foldseek (Van Kempen et al., 2024) clustered version of the AFDB. In addition to this dataset, we use the CATH test set (Ingraham et al., 2019) to assess the reconstruction quality of our VQ-VAE.

**Foldseek Clustered AFDB.** This is the dataset used by Lin et al. (2024) and Geffner et al. (2025b). It is a filtered and clustered rendition of the AlphaFold database. The clustering employs both sequence and structure information (Van Kempen et al., 2024). The resulting dataset is composed of cluster representatives, meaning that one structure is selected from each cluster. This initially yields approximately three million unique samples. They are further filtered following several criteria: a minimum average pLDDT score of 80, protein lengths of less than 256 residues. Recent work (Geffner et al., 2025a) show that using additional secondary structure filters could increase the percentage of $\beta$-sheet content. We thus collect the same secondary structure filtered subset to train AutoFold models. Our results in Section E.3 indicate that using such a subset is indeed helpful for reducing the helix bias in the generated protein structures.

**CATH Dataset.** The CATH dataset consists of experimental structures used by Ingraham et al. (2019). It contains 1140 structures and is used to assess the generalizability of our VQ-VAE models.

## D.2 Computing Details

We implemented our sequence models using the model hub of Hugging Face. Users can easily use their preferred language models to train our autoregressive models instead of using our default Llama model architecture. Experiments were conducted on a shared computing cluster with various CPU and GPU configurations, including 16 NVIDIA H100 (80GB) GPUs. Each of our VQ-VAE models was trained on a single GPU for less than 2 days, and each of our autoregressive models was trained on 4 GPUs for either 2 or 5 days, depending on the model size. Note that, compared to state-of-the-art diffusion or flow matching models, such as La-Proteina, usually requiring 64 NVIDIA A100 (80GB) GPUs, the training of AutoFold requires substantially fewer GPUs.

## D.3 Evaluation Metrics

We evaluate our models using metrics that have become standard in the field. We closely follow the metric definitions given in Geffner et al. (2025b) and Geffner et al. (2025a).

**Designability and Co-Designability** Designability scores are calculated by stripping the generated structure of its sequence and generating 8 sequences using ProteinMPNN Dauparas et al. (2022). These sequences are then refolded with ESMFold Lin et al. (2023). A sample is considered designable if at least one of the refolded structures has an $\alpha$-carbon RMSD below 2Å relative to the generated structure. Designability evaluates whether a sequence folding into the designed structure can be found.
Co-designability, on the other hand, evaluates the self-consistency between the generated sequence and structure. The generated sequence is directly refolded with ESMFold and the backbone RMSD between the structures is computed. If it is below the cutoff of 2Å, the sample is deemed co-designable. Both metrics are reported as fractions of all generated samples produced by a model passing that filter.

**Novelty** Novelty describes how different or new a generated structure is compared to a reference set. We use the PDB set as provided by Foldseek (Van Kempen et al., 2024) and a filtered version of Foldseek's AlphaFold database, as described in Geffner et al. (2025a). TM-scores against the reference set are computed using Foldseek, and the maximum of these TM-scores is reported. A higher TM-score indicates that a similar structure already exists in the reference set. If no similar structures are identified by Foldseek, the TM-score is assigned a value of 0. We then report the average of these maximal scores for all generated proteins passing the co-designability filter.

**Diversity** The diversity assesses the dissimilarity between generated samples of a model. We assess this by computing Foldseek clusters from our generated samples. We consider structural, sequence, and joint (structure and sequence) diversity, using the respective Foldseek clustering. The reported diversity metric is defined as the number of Foldseek clusters formed from co-designable generated proteins, divided by the total number of co-designable proteins generated by a model. The commands used are reproduced below:

Structure diversity:

```
$ foldseek easy-cluster <path_samples> <path_results> <path_tmp>
--cov-mode 0
--alignment-type 1
--min-seq-id 0
--tmscore-threshold 0.5
```

Joint diversity:

```
$ foldseek easy-cluster <path_samples> <path_results> <path_tmp>
--cov-mode 0
--alignment-type 2
--min-seq-id 0.1
--tmscore-threshold 0.5
```

Sequence diversity:

```
$ mmseqs easy-cluster <fasta_input_filepath>
pdb_cluster <path_tmp>
--min-seq-id 0.1
-c 0.7
--cov-mode 1
```

**Unique Co-Designability** We introduce a new composite metric that we call unique co-designability. Unique co-designability is the multiplication of structural diversity by the co-designability score. We believe it to be a very relevant metric since it illustrates how many different co-designable proteins a model can produce, whereas co-designability could be very high for a model producing only close to identical proteins.

**Sequence FID** We also introduce a new metric to measure the distribution discrepancy between the training, here AFDB, and the generated sequences. We first compute sequence embeddings using the `esm2_t33_650M_UR50D` model provided by ESM2 (Lin et al., 2023). Then, we compute the Fréchet distance between the embeddings of the generated and training sequences from AFDB. Lower sequence FID indicates that the generated sequences are more similar to the training sequences.

### D.4 HYPERPARAMETERS

Here, we provide a full set of hyperparameters used to train the VQ-VAE and the autoregressive models.

**VQ-VAE.** Our choices of the hyperparameters for VQ-VAE are largely based on those of ESM-3 (Hayes et al., 2025). Table 3 provides full details about the hyperparameters used to train the VQ-VAE models.

**AutoFold.** Following standard choices of training Llama models (Touvron et al., 2023a;b), we train AutoFold with different parameter sizes ranging from 27M to 407M. Our choice of the hyperparameters strictly follows the standard choices for training large language models without specific tuning. Table 4 summarizes the full details of the training hyperparameters for AutoFold models. For inference, we adopt the top-$p$ sampling with annealed temperature. As we have different types of tokens in our sequence representation of the graph, we use different temperatures when generating different types. All these values for different models tuned with different aims are provided in Table 5.

### D.5 MOTIF SCAFFOLDING

Given a certain selection of residues' information about backbone position and their amino acids, the task of motif scaffolding is to generate a new protein that includes this motif as part of it. As an important distinction from previous models, our AutoFold models trained for unconditional generation can be seamlessly used for motif scaffolding, without any fine-tuning. We provide details below about the benchmark datasets as well as the evaluation.

**Benchmark Datasets.** We use the same benchmark datasets as used by previous work (Watson et al., 2023; Geffner et al., 2025a), excluding sparse active site motifs. Table 6 includes all benchmark data used in our experiments.

**Evaluation.** Following La-Proteina (Geffner et al., 2025a) and the fact that our model only generates the backbone structure, we assess the validity of the generated samples through the following criteria:

- The motif backbone coordinates should have an RMSD $< 1$ Å between the generated and input structures.
- The generated protein should be co-designable, i.e., it should have a backbone scRMSD $< 2$ Å.

Table 3: Training hyperparameters for VQ-VAE

| Hyperparameter name | Value |
|---|---|
| *Architecture* | |
| # Parameters | 218M |
| Hidden dimension | 1024 |
| # Geometric encoder blocks | 2 |
| # Anchor heads | 128 |
| # RBF kernels | 16 |
| Contact distance threshold | 8.0 |
| Codebook dimension | 128 |
| Codebook size | 256 |
| Codebook EMA decay | 0.99 |
| Codebook commitment coefficient | 0.25 |
| # Decoder MPNN blocks | 4 |
| # Decoder transformer blocks | 4 |
| # Heads in transformer blocks | 16 |
| *Optimization* | |
| Batch size $\times$ accumulate gradient batches | $16 \times 4$ |
| Learning rate | 0.0004 |
| Weight decay | 0.01 |
| Betas | (0.9, 0.95) |
| Warm up ratio | 0.05 |
| Gradient clipping value | 1.0 |
| Training iterations | 100,000 |

Table 4: Training hyperparameters for AutoFold models

| Hyperparameter name | AutoFold-t | AutoFold-s | AutoFold-m |
|---|---|---|---|
| *Architecture* | | | |
| # Parameters | 27M | 116M | 407M |
| Hidden dimension | 512 | 768 | 1024 |
| # Transformer blocks | 6 | 12 | 24 |
| *Optimization* | | | |
| Batch size $\times$ accumulate gradient batches | $128 \times 2$ | $64 \times 4$ | $32 \times 8$ |
| Truncation length | | 5120 | |
| Learning rate | 0.0006 | 0.0006 | 0.0003 |
| Weight decay | | 0.1 | |
| Betas | | (0.9, 0.95) | |
| Warm up ratio | | 0.01 | |
| Gradient clipping value | | 1.0 | |
| Training iterations | | 100,000 | |

Table 5: Inference hyperparameters for AutoFold

| Hyperparameter name | Value |
| --- | --- |
| *Unconditional generation* | |
| Tuned to optimize co-designability | |
| Top-$p$ sampling | 0.7 |
| Graph token temperature | 0.2 |
| Sequence token temperature | 1.0 |
| Structure token temperature | 0.4 |
| *Unconditional generation* | |
| Tuned to maximize beta-sheet content | |
| Top-$p$ sampling | 0.6 |
| Graph token temperature | 0.9 |
| Sequence token temperature | 0.9 |
| Structure token temperature | 0.9 |
| *Motif Scaffolding* | |
| Top-$p$ sampling | 0.7 |
| Graph token temperature | {0.5, 0.1} |
| Sequence token temperature | {0.5, 0.1} |
| Structure token temperature | {0.5, 0.1} |

Table 6: Motif scaffolding data with minimum and maximum lengths, and contig strings.

| Motif Name (PDB ID) | Min Length | Max Length | Contig String All Atom |
| --- | --- | --- | --- |
| 1YCR | 40 | 100 | 10-40/B19-27/10-40 |
| 1PRW | 60 | 105 | 5-20/A1-20/10-25/B1-20/5-20 |
| 1BCF | 96 | 152 | 8-15/A92-99/16-30/A123-130/16-30/A47-54/16-30/A18-25/8-15 |
| 5TPN | 39 | 98 | 10-40/A163-181/10-40 |
| 5IUS | 57 | 142 | 0-30/A119-140/15-40/A63-82/0-30 |
| 3IXT | 50 | 75 | 10-40/P254-277/10-40 |
| 4JHW | 59 | 104 | 10-25/F196-212/15-30/F63-69/10-25 |
| 5WN9 | 40 | 100 | 10-40/A170-189/10-40 |
| 4ZYP | 35 | 95 | 10-40/A422-436/10-40 |
| 6VW1 | 62 | 83 | 20-30/A24-42/4-10/A64-82/0-5 |
| 2KL8 | 79 | 79 | A1-7/20/A28-79 |
| 7MRX_60 | 60 | 60 | 0-38/B25-46/0-38 |
| 7MRX_85 | 85 | 85 | 0-63/B25-46/0-63 |
| 7MRX_128 | 128 | 128 | 0-122/B25-46/0-122 |
| 5TRV_short | 56 | 56 | 0-35/A45-65/0-35 |
| 5TRV_med | 86 | 86 | 0-65/A45-65/0-65 |
| 5TRV_long | 116 | 116 | 0-95/A45-65/0-95 |
| 6E6R_short | 48 | 48 | 0-35/A23-35/0-35 |
| 6E6R_med | 78 | 78 | 0-65/A23-35/0-65 |
| 6E6R_long | 108 | 108 | 0-95/A23-35/0-95 |

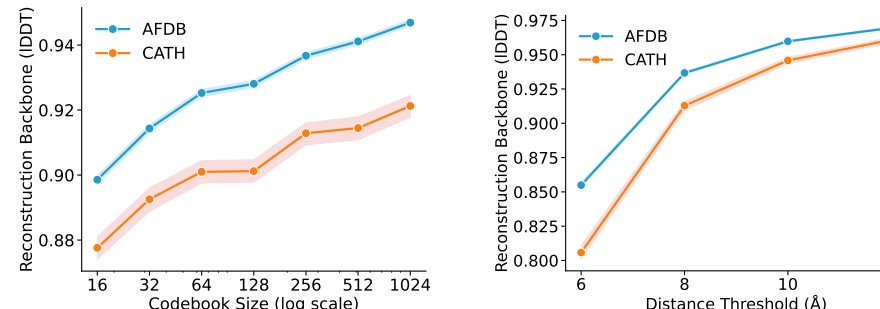

Figure 5: Backbone reconstruction lDDT for VQ-VAE models. Left: reconstruction lDDTs vs. the codebook size in VQ-VAE. Right: reconstruction lDDTs vs. the distance threshold used to construct the graphs. 95% confidence intervals are shown by shading.

For all methods, we generate 200 samples per task. We then evaluate these samples via the criteria above, which results in the number of successes per task. Finally, the number of unique successes is obtained by clustering the successes with Foldseek (Van Kempen et al., 2024) and reporting the number of clusters. We use the following command to cluster:

```
$ foldseek easy-cluster <path_samples> <path_tmp>/res <path_tmp>
--alignment-type 1 --cov-mode 0 --min-seq-id 0
--tmscore-threshold 0.5 --single-step-clustering
```

## E    ADDITIONAL RESULTS

We present additional results for our VQ-VAE models and AutoFold models.

### E.1    ADDITIONAL RESULTS FOR VQ-VAE

To further supplement the reconstruction results in Section 5.1, we provide the reconstruction lDDT scores in Figure 5. Our lDDT results are consistent with the RMSD results.

### E.2    PAE AND PLDDT COMPARISON TO LA-PROTEINA

We compare the distribution of pAE and pLDDT scores from Autofold-m and La-Proteina in Figure 6. Note that for La-Proteina 100 proteins for lengths 100, 200, 300, 400, and 500 each are assessed. Our results show that AutoFold-m is only slightly outperformed by La-Proteina, despite being a fully discrete autoregressive model.

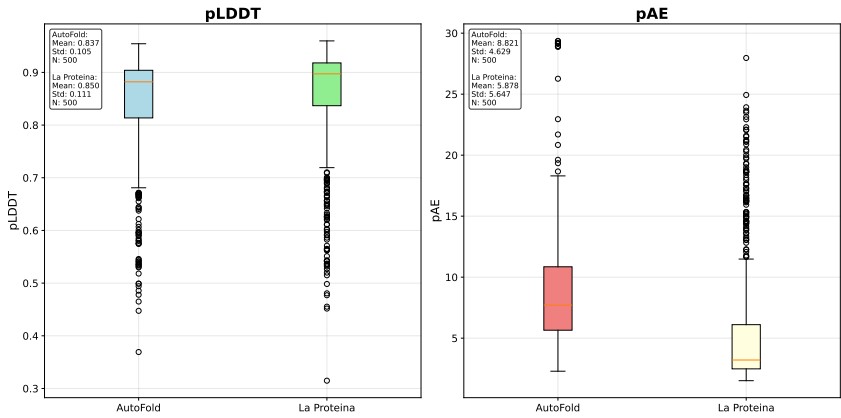

Figure 6: Comparison of pAE and pLDDT distributions between Autofold-m and La-Proteina

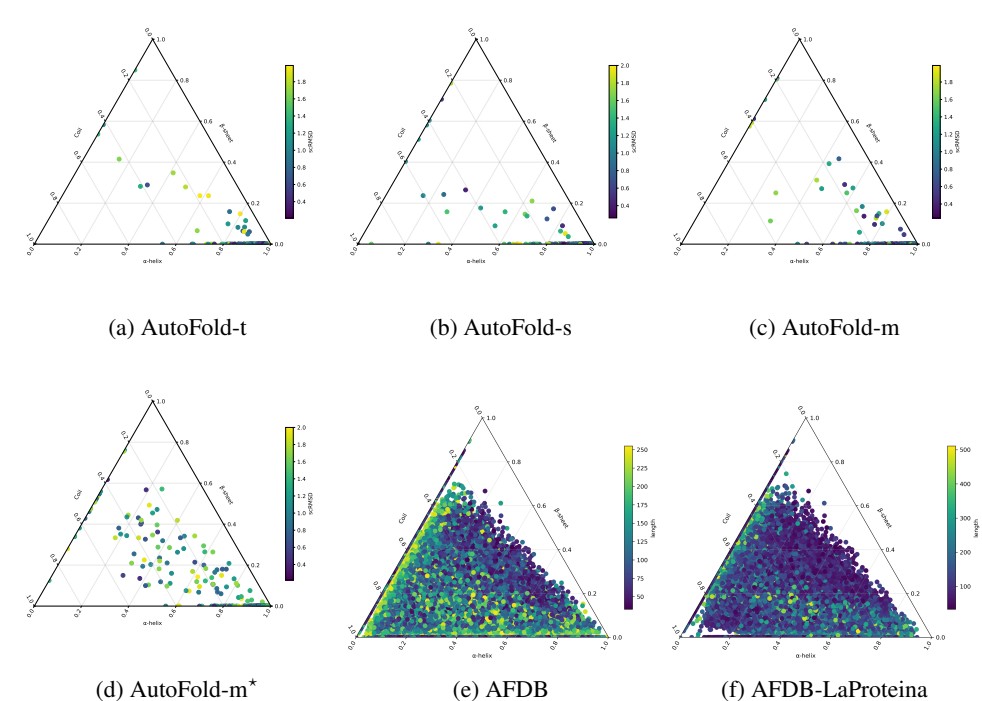

(a) AutoFold-t     (b) AutoFold-s     (c) AutoFold-m

(d) AutoFold-m$^\star$     (e) AFDB     (f) AFDB-LaProteina

Figure 7: Distribution of secondary structure elements ($\alpha$-helices, $\beta$-sheets, and coils) of co-designable proteins generated by different models. The inference hyperparameters for all AutoFold models were selected to maximize the $\beta$-sheet content. AutoFold-m$^\star$ generates a good balance of helices and sheets.

### E.3 DISTRIBUTION OF SECONDARY STRUCTURE ELEMENTS

We conduct a secondary structure composition analysis of the generated protein structures, similar to FoldFlow2 (Huguet et al., 2024). Our results in Figure 7 demonstrate that AutoFold-m$^\star$, trained on the secondary-structure-filtered AFDB dataset used by La-Proteina (Geffner et al., 2025a), achieves a good balance between helices and sheets. However, the same model trained on AFDB achieves much lower $\beta$-sheet content. This suggests that the choice of the training dataset is crucial for generative models to achieve a less biased $\alpha$-helix/$\beta$-sheet proportion.

### E.4 LENGTH DISTRIBUTION OF GENERATED PROTEINS

We provide length distributions of co-designable proteins generated by different AutoFold models in Figure 8. Our results indicate that the proteins generated by AutoFold largely match the training length distribution, with a modest shift toward shorter proteins.

## F VISUALIZATION OF GENERATED STRUCTURES

### F.1 UNCONDITIONAL GENERATION

We provide examples of generated structures for unconditional generation in Figure 9, 10, 11, 12, and 13 respectively for AutoFold-m (tuned to optimized co-designability), AutoFold-m, AutoFold-s, AutoFold-t, and AutoFold-m$^\star$ (tuned to maximize $\beta$-sheet content).

### F.2 MOTIF SCAFFOLDING

We provide several examples of generated structures in Figure 14.

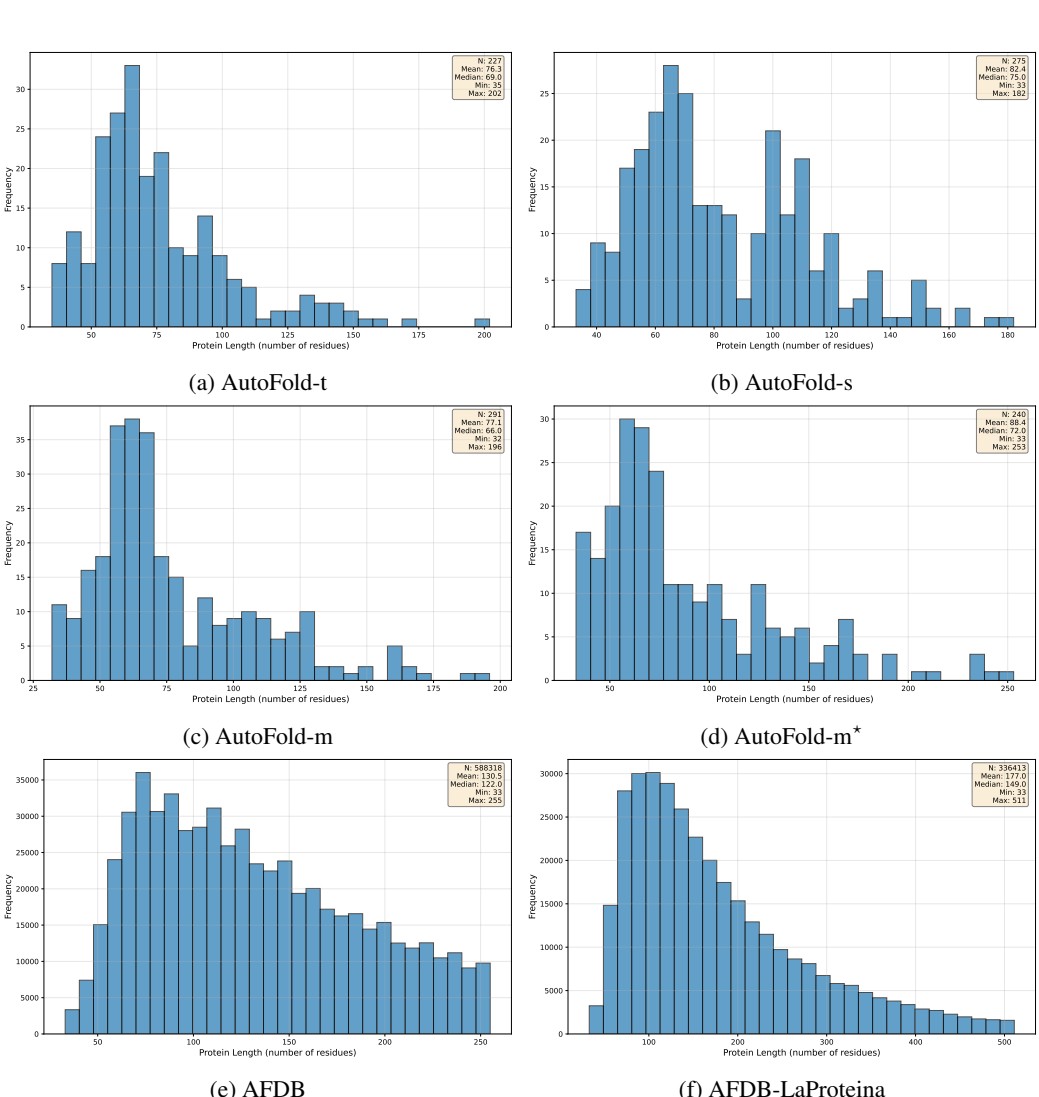

Figure 8: Length distribution of co-designable proteins generated by different AutoFold models. The proteins generated by AutoFold largely match the training length distribution, with a modest shift toward shorter proteins.

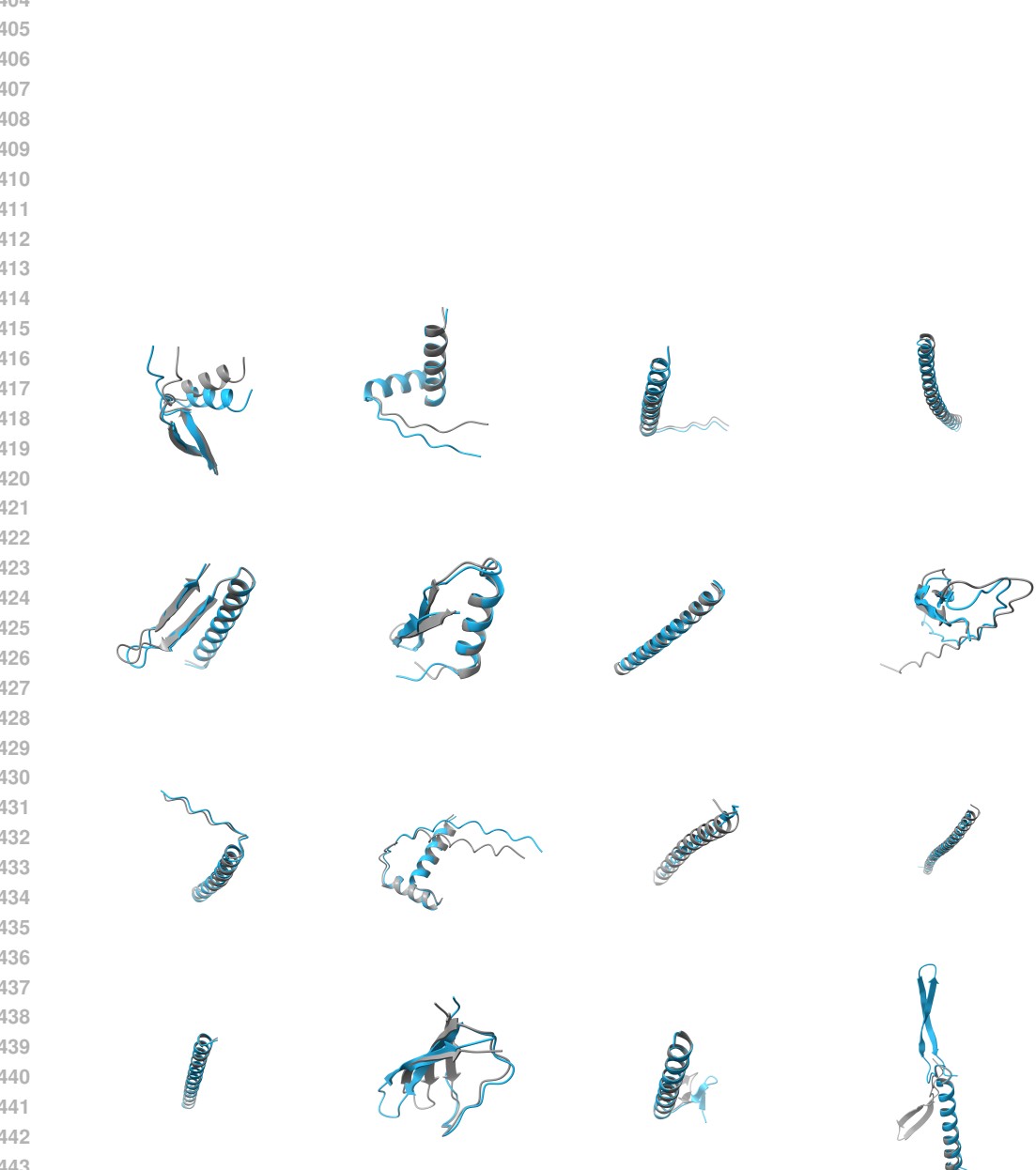

Figure 9: Further examples of generated structures by AutoFold-m (with inference hyperparameters selected to maximize co-designability) showcasing an overabundance of $\alpha$-helices. Blue: AutoFold generated structures. Gray: ESMFold predicted structures. Note the last structure as an example of a structure considered non-designable.

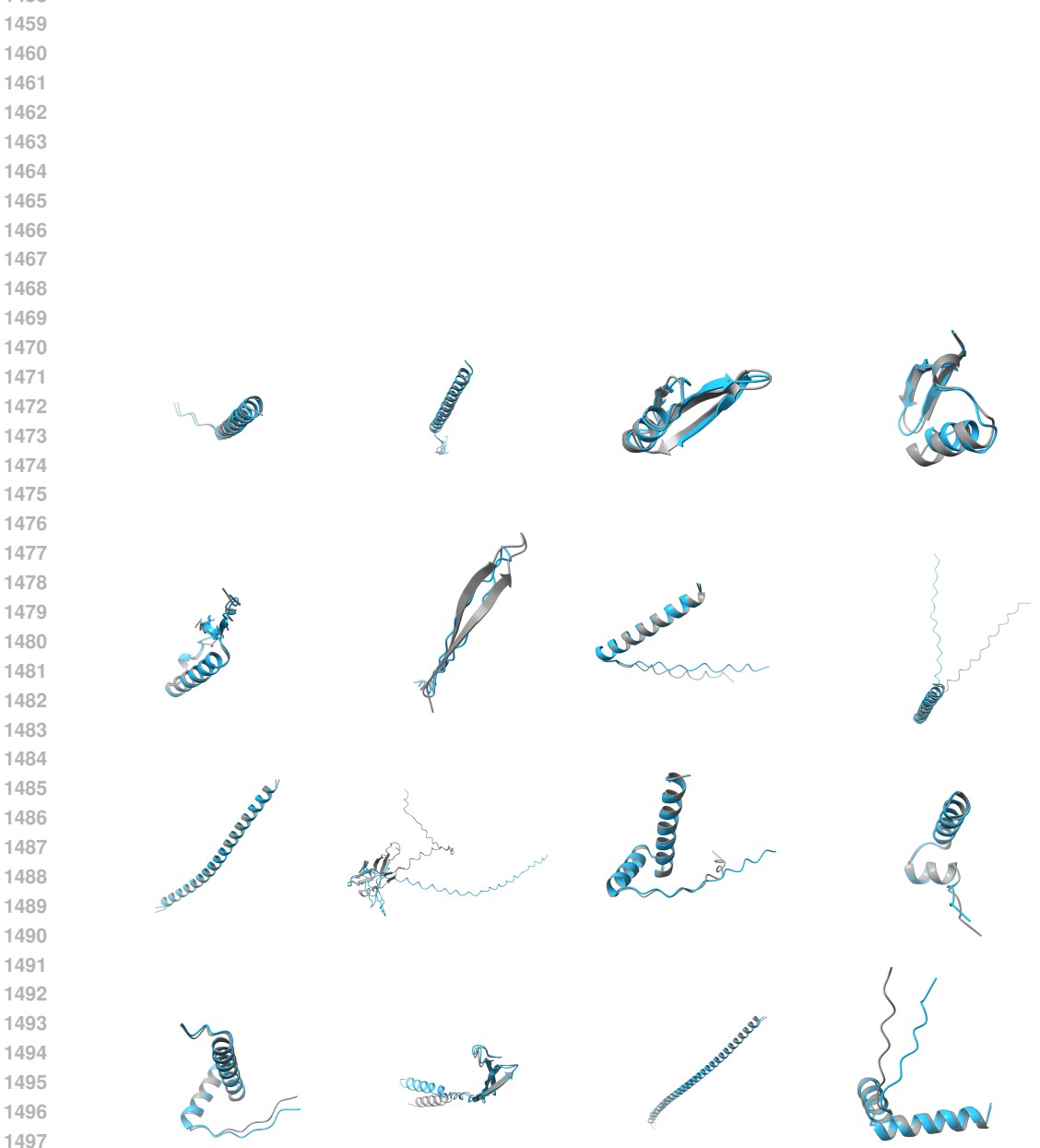

Figure 10: Examples of AutoFold-m with inference hyperparameters selected to maximize $\beta$-sheet content. Blue: AutoFold generated structures. Gray: ESMFold predicted structures.

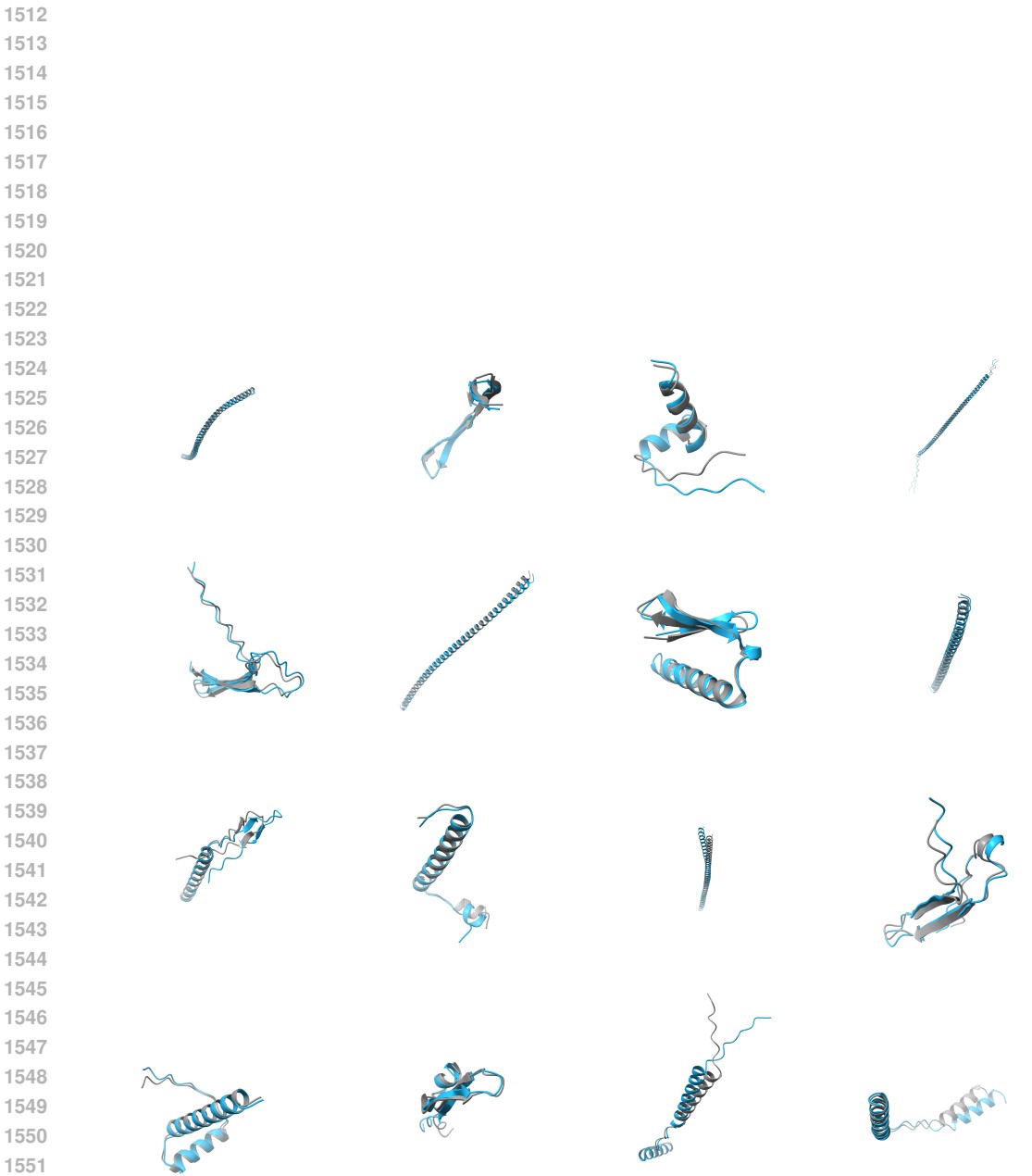

Figure 11: Examples of AutoFold-s with inference hyperparameters selected to maximize $\beta$-sheet content. Blue: AutoFold generated structures. Gray: ESMFold predicted structures.

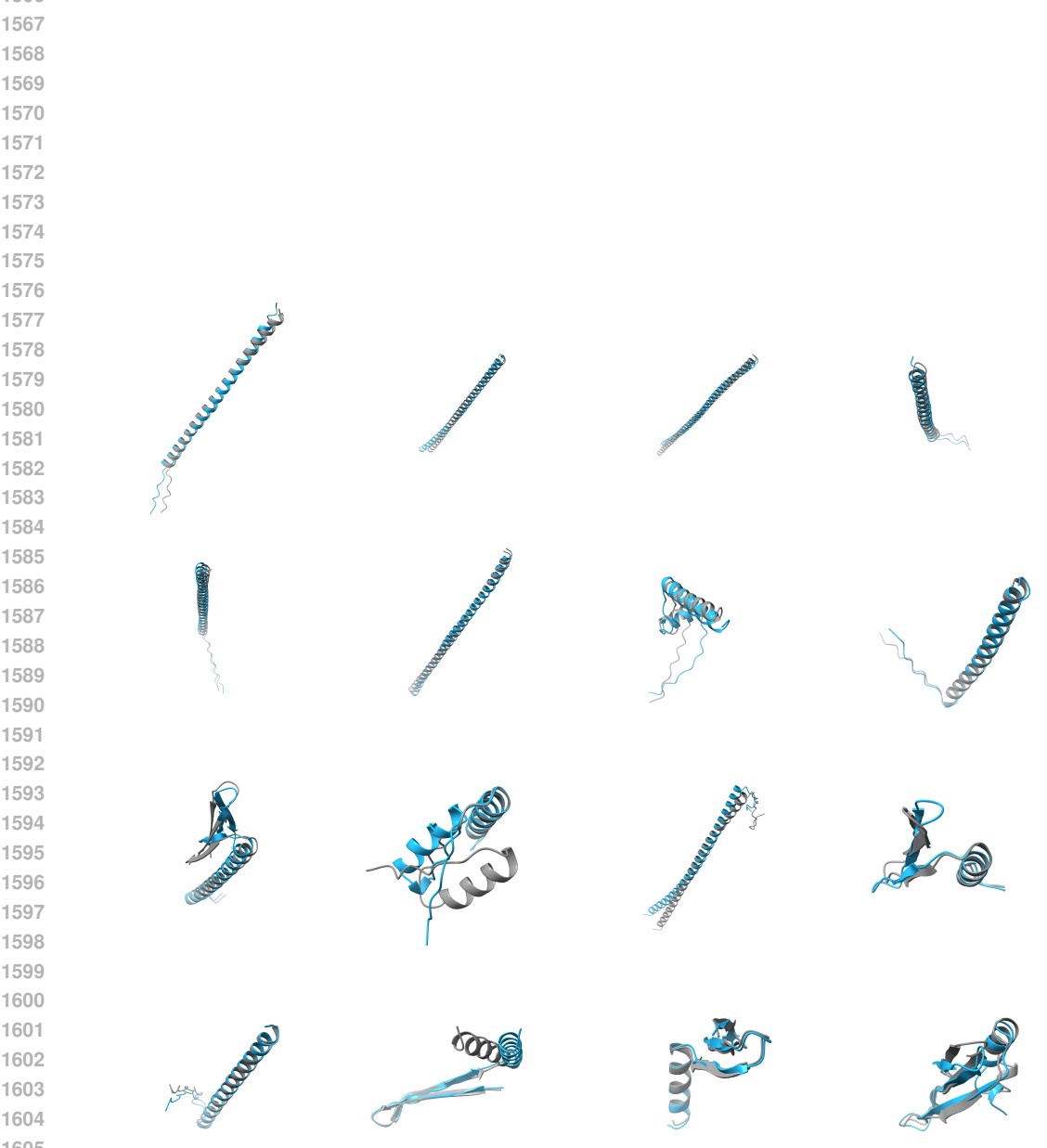

Figure 12: Examples of AutoFold-t with inference hyperparameters selected to maximize $\beta$-sheet content. Blue: AutoFold generated structures. Gray: ESMFold predicted structures.

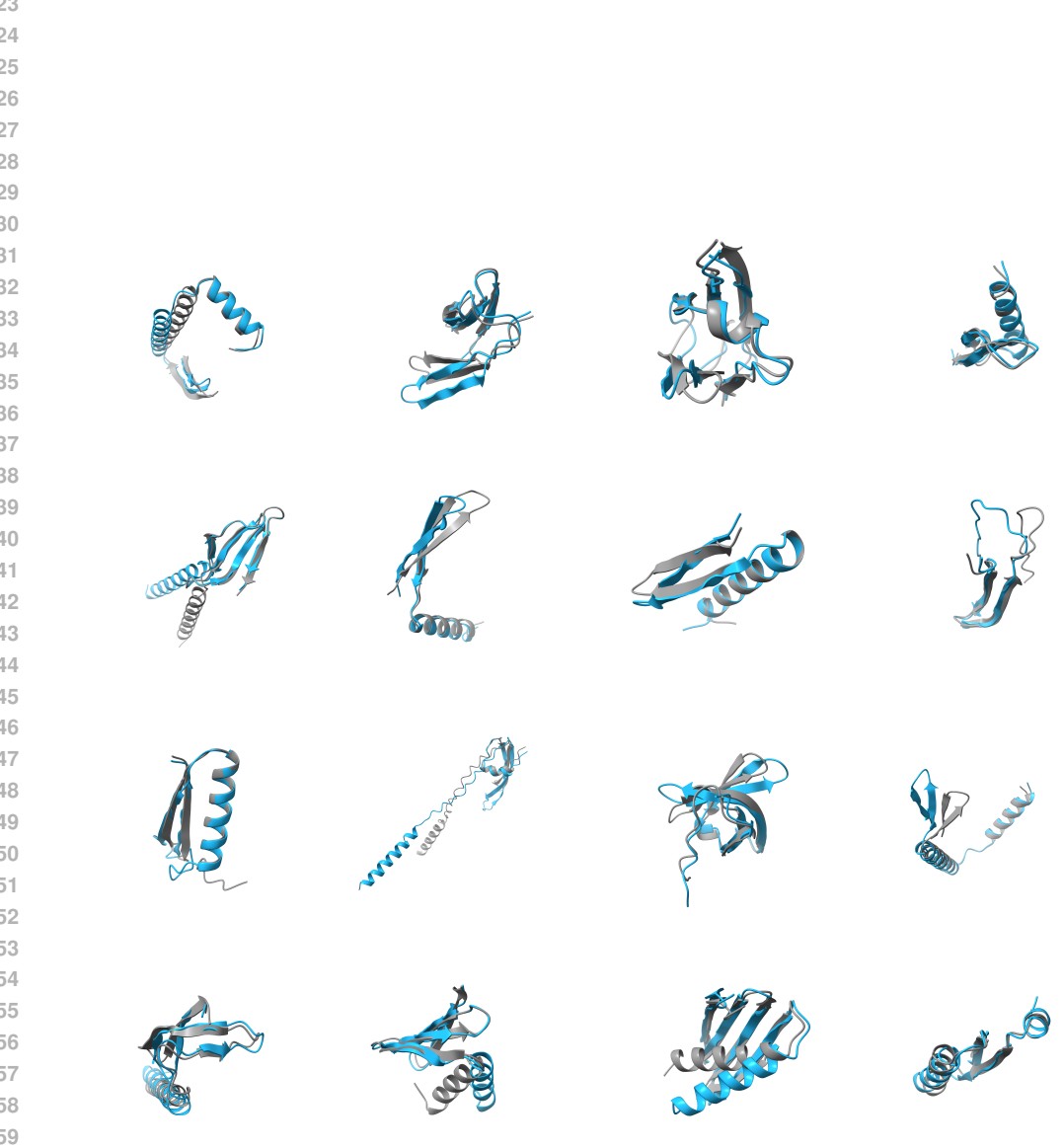

Figure 13: Examples of AutoFold-m$^\star$ trained on AFDB-LaProteina with inference hyperparameters selected to maximize $\beta$-sheet content. Blue: AutoFold generated structures. Gray: ESMFold predicted structures.

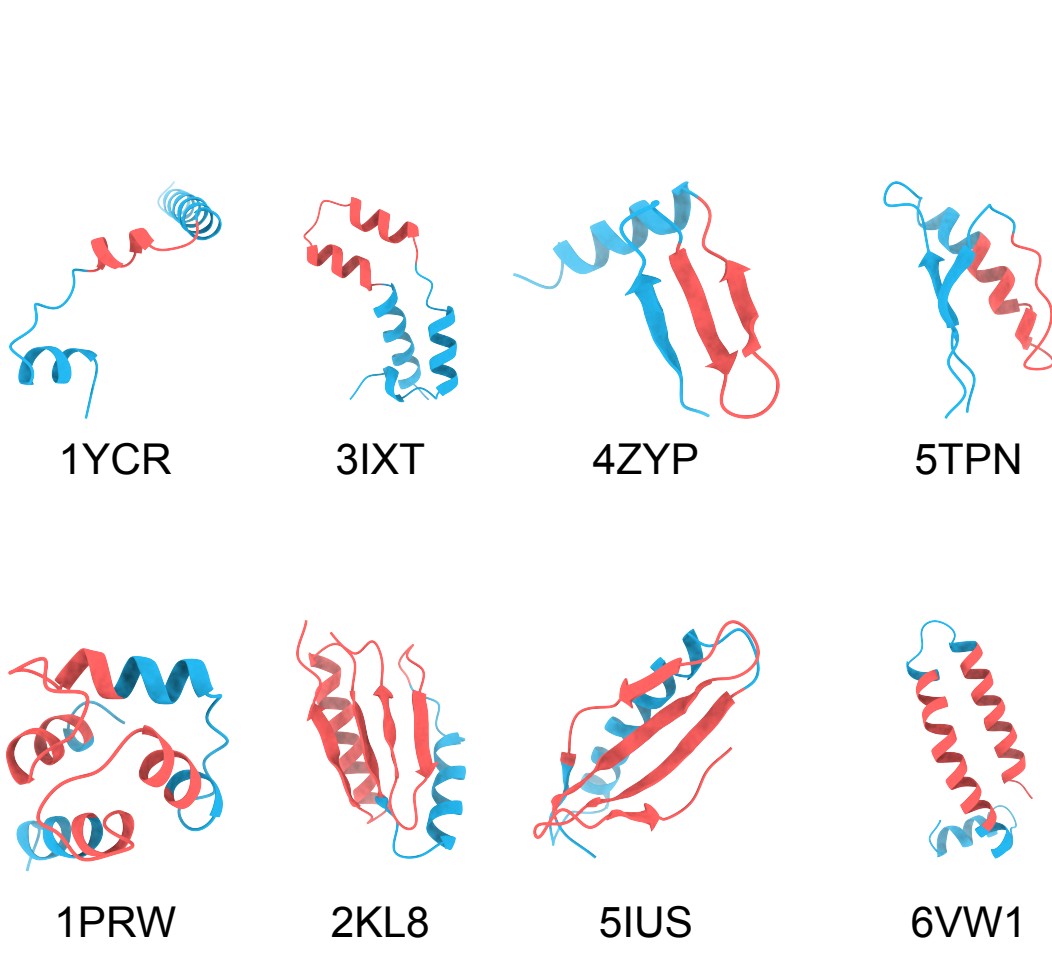

Figure 14: Examples of generated structures for motif scaffolding tasks. The red part indicates the motif structures.

Table 7: Reconstruction quality comparison of a VQ-VAE model with and without edge symmetrization. 95% confidence intervals are shown in the brackets.

| Symmetrization | AFDB | | CATH | |
|---|---|---|---|---|
| | RMSD ↓ | lDDT ↑ | RMSD ↓ | lDDT ↑ |
| w/ | **0.807**([0.791, 0.826]) | **0.937**([0.936, 0.938]) | 0.676([0.636, 0.721]) | **0.913**([0.909, 0.916]) |
| w/o | 0.812([0.795, 0.831]) | 0.933([0.932, 0.933]) | 0.676([0.635, 0.725]) | 0.910([0.906, 0.913]) |

# G  ABLATION EXPERIMENTS

Here, we provide several ablation experiments to support the choices made in our main experiments.

## G.1  IMPACT OF EDGE SYMMETRIZATION

We first study the impact of the symmetrization of edge features. Without symmetrization, our VQ-VAE would generate a bidirected graph, making the generation task more challenging for autoregressive models. We show in Table 7 that this choice of symmetrizing edges does not degrade but even slightly improves the reconstruction quality of the VQ-VAE models.

## G.2  IMPACT OF MODEL SIZE

In addition to the results of model sizes for unconditional generation presented in Section 5.2, we investigate the impact of model size on AutoFold for motif scaffolding. Table 8 presents the number of successful designs for all motifs. Our results demonstrate that AutoFold's performance increases monotonically with the model size, as indicated by the number of successful designs. The largest model, AutoFold-m, performs the best across almost all tasks. This suggests that increasing the model size may further improve the performance.

1782
1783
1784
1785
1786
1787
1788
1789
1790
1791
1792
1793
1794
1795
1796
1797
1798
1799
1800
1801
1802
1803
1804
1805
1806
1807
1808
1809
1810
1811
1812
1813
1814
1815
1816
1817
1818
1819
1820
1821
1822
1823
1824
1825
1826
1827
1828
1829
1830
1831
1832
1833
1834
1835

Table 8: Impact of the model size on motif scaffolding tasks

| Motif | # Segments | AutoFold-t | | AutoFold-s | | AutoFold-m | |
|---|---|---|---|---|---|---|---|
| | | All | Unique | All | Unique | All | Unique |
| 1YCR | 1 | 67 | 32 | 52 | 37 | 68 | **44** |
| 3IXT | 1 | 2 | 1 | 4 | 2 | 11 | **6** |
| 4ZYP | 1 | 3 | 2 | 17 | **8** | 23 | **8** |
| 5TPN | 1 | 2 | 2 | 4 | **4** | 1 | 1 |
| 5WN9 | 1 | 0 | 0 | 0 | 0 | 2 | **1** |
| 5TRV_short | 1 | 1 | 1 | 3 | 2 | 4 | **3** |
| 5TRV_med | 1 | 0 | 0 | 0 | 0 | 0 | 0 |
| 5TRV_long | 1 | 0 | 0 | 0 | 0 | 0 | 0 |
| 6E6R_short | 1 | 10 | 6 | 12 | 8 | 23 | **16** |
| 6E6R_med | 1 | 0 | 0 | 3 | 1 | 8 | **4** |
| 6E6R_long | 1 | 0 | 0 | 2 | **2** | 1 | 1 |
| 7MRX_60 | 1 | 6 | 2 | 7 | 2 | 20 | **5** |
| 7MRX_85 | 1 | 3 | **1** | 2 | **1** | 8 | **1** |
| 7MRX_128 | 1 | 0 | 0 | 3 | **1** | 2 | **1** |
| 1PRW | 2 | 50 | 7 | 141 | **20** | 162 | 16 |
| 2KL8 | 2 | 161 | **5** | 197 | 3 | 200 | 1 |
| 4JHW | 2 | 0 | 0 | 0 | 0 | 1 | **1** |
| 5IUS | 2 | 2 | 2 | 28 | 7 | 52 | **9** |
| 6VW1 | 2 | 89 | 4 | 75 | 5 | 86 | **6** |
| 1BCF | 4 | 6 | 1 | 88 | **10** | 157 | 9 |
| Total | | 398 | 64 | 638 | 113 | **838** | **133** |
| Total (w/o fixed length) | | 378 | 54 | 606 | 96 | **772** | **102** |