# OpenReview forum: "AutoFold: Ultra-fast Protein Generation via Autoregressive Contact Graph Generation"
_ICLR.cc/2026/Conference — Submitted to ICLR 2026_

### Official Review · Reviewer_pLGR · 2025-10-30

**Soundness:** 3
**Presentation:** 4
**Contribution:** 3
**Rating:** 6
**Confidence:** 4

**Summary:**

The paper introduces AutoFlow, a model for co-design of protein backbones and sequences. The main idea behind the model is to train an autoencoder for protein backbones and sequences, and perform generation autoregressively in the latent space of the autoencoder. By doing this, the model can enforce SE(3) equivariance, and generates proteins with lower inference times than methods that work on 3D space. The authors show results that are competitive with other existing methods.

**Strengths:**

**Originality**: The idea of doing co-design in the latent space of an autoencoder, instead of in 3D space, is a welcome change from the large and ever increasing amount of models that run a generative model on atomic coordinates.

**Quality**: The model performance is comparable to that of the models they compare against (La Proteina, Protpardelle, etc.) despite the lower generation time

**Clarity**: The paper is very well written and presented. The introduction does a good review of the existing literature. The model is described clearly, and figure 1 is a great aid. Results figures and tables are well laid out. The honesty about the weaknesses of the model (e.g. producing too many alpha helices) is very appreciated.

**Weaknesses:**

- Perhaps not a weakness, as much as something that needs to be clarified, but it is not obvious to me what the advantage of faster inference time is for protein design (though I admit there could be advantages I am not aware of). What I mean is, when it comes to protein folding, for example, faster inference means that one could do virtual screening. However (and again, I am ready to be corrected here), my feeling is that if I was going to do in silico design of a protein to perform some function, inference time is not really a bottleneck (especially compared to other testing and trials the designed protein would need to undergo to be useful), as long as the protein I design folds correctly and does what I want it to do. Of course, less inference time means one could scale design to bigger proteins, but aside from that, it is unclear to me what the advantage of the faster inference time would be for practical drug design. Once more, I would like to stress that there are probably applications I am not thinking about here, I would just like to encourage the authors to add some more focus on why this is something that helps drug design.
- Somewhat related to the point above, but the motivation for the choice of codebook size and distance threshold for the autoencoder is not at all clear. Figure 2 clearly shows that (as expected) larger numbers lead to better performance. Therefore, it would be good to know if the reason is that these settings reach some threshold memory usage, or inference time. Otherwise, why not use settings that lead to better performance?
- It would have been good to see ca omparison in designability to structure-only models (e.g. RFDiffusion, Foldflow) and also a comparison in sequence quality to sequence only models (e.g. ESM).
- Some of the results require more detailed explanation (see questions below)

**Questions:**

- In section 4.3, the authors discuss potential applications in unconditional design and motif scaffolding. Could the approach used for motif scaffolding also be used for binder design? It would be good to add a comment on this.
- In section 5, experimental setup, it is not clear how the split is done. Is it using mmseqs clustering, for example?
- In the same section, why is the model trained on alphafold structures only, and no PDB structures are included?
- In section 5.1, it would be good to understand why the performance on CATH (which should in principle be more out of distribution) is seemingly better than on AFDB
- In the same section, in figure 2, it would be good to see for which values in the x-axis, the performance plateaus (it seems to continue to get better for the values shown).
- Another comment to that section and figure, if that is possible, it would be much more useful to view a 2D grid of performance as a function of distance threshold and codebook.
- Related to a point brought up in the weaknesses section, but the motivation to settle for codebook size 256 and distance threshold 8 is not at all explained.
- In section 5.2, sequence quality is evaluated through FID of ESM2 embeddings. There exist alternatives to FID that claim to be better (such as Feature Likelihood Divergence, PQMass) it would be interesting to see results on those as well if possible
- In table 1, the smaller Autofold seems to do better in some categories. It would be good to get an explanation for this behaviour.
- Similarly, there is no comment about the big gap in designability compared to La Proteina
- There is also no discussion (as far as I can see) about the choice of parameters for generation, such as temperature, and how they affect these results.
In table 2, the results for AutoFold are much better with a distance threshold of 9. This once again raises the question of why 8 was chosen.

---

> ### Author Response · Authors · 2025-11-24
> **Author response to Reviewer pLGR**
>
> We thank you for your thoughtful and constructive feedback. Please find our response to each of your points below.
>
> > W1 Perhaps not a weakness, as much as something that needs to be clarified, but it is not obvious to me what the advantage of faster inference time is for protein design (though I admit there could be advantages I am not aware of). What I mean is, when it comes to protein folding, for example, faster inference means that one could do virtual screening. However (and again, I am ready to be corrected here), my feeling is that if I was going to do in silico design of a protein to perform some function, inference time is not really a bottleneck (especially compared to other testing and trials the designed protein would need to undergo to be useful), as long as the protein I design folds correctly and does what I want it to do. Of course, less inference time means one could scale design to bigger proteins, but aside from that, it is unclear to me what the advantage of the faster inference time would be for practical drug design. Once more, I would like to stress that there are probably applications I am not thinking about here; I would just like to encourage the authors to add some more focus on why this is something that helps drug design.
>
> We agree that, for many protein-design pipelines, inference speed is not the primary bottleneck compared to downstream filtering, experimental validation, and functional testing. We did not intend to suggest that speed alone is the main value of AutoFold. Rather, we view efficiency as a useful property given a fixed compute or time budget. Faster generation can make quicker iterations feasible and eventually compensate for lower co-designability. As this is hard to demonstrate in our experiments, we have removed “ultra-fast” from the title and revised the introduction to emphasize the methodological focus.
>
> In addition to this positioning clarification, we want to highlight that our main contribution is the introduction of a compact, sparse, discrete contact-graph representation of proteins that is suitable for generative modeling. This representation provides a new perspective on protein structure with practical implications beyond raw speed:
>
> - **Efficiency and scalability on sparse structure graphs.** Working on sparse contact graphs is substantially cheaper than operating on dense residue-pair representations, typically used in diffusion/flow matching models.
> - **A single model for both unconditional and substructure-conditional generation.** The autoregressive graph formulation supports motif scaffolding by prompting substructures (motifs), without retraining or fine-tuning. This unified capability is a key practical advantage for real design tasks.
> - **Token-based modeling that aligns with multimodal LLM paradigms.** The discrete graph tokens provide a natural interface to language-modeling machinery. This opens the door to multimodal protein LLMs that can jointly reason over sequence and (sub)structure in a unified token space, and to flexible conditional generation via text prompting.
>
> > W2 Somewhat related to the point above, but the motivation for the choice of codebook size and distance threshold for the autoencoder is not at all clear. Figure 2 clearly shows that (as expected) larger numbers lead to better performance. Therefore, it would be good to know if the reason is that these settings reach some threshold memory usage or inference time. Otherwise, why not use settings that lead to better performance?
>
> Our choices of the codebook size 256 and distance cutoff were not driven by hardware limits or inference-time constraints. Instead, these hyperparameters were guided by methodological considerations tied to the goals of this paper:
>
> - Our primary goal is to test whether protein structures admit a _usefully sparse_ contact-graph representation that can be generated through graph generation models (in our work, an autoregressive model), rather than directly modeling continuous 3D coordinates. A cutoff of 8.0 Å is a common choice for the contact map [1] primarily because a majority of local intermolecular interactions manifest within this range [2].
> - We fix the codebook size at 256 to maintain a good balance between reconstruction quality and generation difficulty. While increasing the vocab size improves the reconstruction quality, it also makes the generative task harder: for example, for the motif scaffolding task, the same motif might be encoded into a graph with different edge labels due to an overused number of codes. And a potential consequence could be that you need more data to get sufficiently frequent subgraph coverage.
>
> We have added a discussion in Section 5.1 of our revised manuscript.

---

> > ### Author Response · Authors · 2025-11-24
> > **Author response to Reviewer pLGR (cont.)**
> >
> > > W3 It would have been good to see a comparison in designability to structure-only models (e.g. RFDiffusion, Foldflow) and also a comparison in sequence quality to sequence only models (e.g. ESM).
> >
> > Thank you for your suggestion. However, we believe that adding such comparisons here would be difficult to interpret fairly, because the objectives and evaluation pipelines are fundamentally different.
> >
> > - Methods like RFDiffusion or FoldFlow generate only backbones and must be paired with a separate inverse-folding model (typically ProteinMPNN) to obtain sequences and evaluate designability. The resulting designability score thus reflects a two-stage pipeline: backbone generation quality and inverse folding quality. ProteinMPNN is trained on native-like structures and can struggle on highly novel backbones, which may eventually depress designability even when the backbone itself is physically plausible. AutoFold, by contrast, jointly generates backbone and sequence, so its co-designability reflects the quality of a single integrated co-design model. Comparing to structure-only conditional benchmarks with co-design conditional benchmarks makes it harder to interpret differences and can lead to misleading conclusions.
> > - Large sequence models such as ESM3 are trained on orders of magnitude more data and are not designed to generate paired structure–sequence samples. Comparing our sequence quality to a sequence-only LM would therefore not be apples-to-apples in either training scale or task definition. Moreover, ESM3 is not a sequence-only model; it is multimodal and was already benchmarked in Proteina, where its designability is surprisingly low.
> >
> > > Q1 In section 4.3, the authors discuss potential applications in unconditional design and motif scaffolding. Could the approach used for motif scaffolding also be used for binder design? It would be good to add a comment on this.
> >
> > In principle, binder design can be cast in the same framework as motif scaffolding. A binder task typically specifies an interface “motif” that must be preserved, and the remaining scaffold is generated around it. This matches our conditional generation mechanism: we can flatten the fixed interface substructure into a prompt and autoregressively generate the rest of the contact graph and sequence.
> >
> > The main additional consideration is multi-chain structure. To represent protein–protein complexes, the contact graph and its SENT tokenization would need to distinguish chains and allow cross-chain edges. This can be handled by introducing new special tokens for chain splitting. This was already briefly discussed in Section A. We are happy to expand it if needed.
> >
> > > Q2 In section 5, experimental setup, it is not clear how the split is done. Is it using mmseqs clustering, for example?
> >
> > We performed a simple random split of our dataset into training and validation sets.
> > For novelty evaluation, we follow the same reference sets and pipeline as La-Proteina: novelty is computed against their PDB/AFDB reference datasets to ensure comparability. We have incorporated this clarification in the revision.
> >
> > > Q3 In the same section, why is the model trained on alphafold structures only, and no PDB structures are included?
> >
> > We trained on AFDB and AFDB-LaProteina structures because we followed the exact dataset and preprocessing pipeline established by prior work (Genie2, Proteina, and La-Proteina). Using the same training source ensures our comparisons to La-Proteina are as controlled as possible. We note that our model can straightforwardly be fine-tuned (or jointly trained) on PDB data if a particular application requires it.
> >
> > > Q4 In section 5.1, it would be good to understand why the performance on CATH (which should in principle be more out of distribution) is seemingly better than on AFDB
> >
> > We hypothesize that our AFDB dataset includes a much wider variety of proteins, including regions with lower confidence. This heterogeneity in topology and quality makes the reconstruction harder.

---

> > > ### Author Response · Authors · 2025-11-24
> > > **Author response to Reviewer pLGR (cont.)**
> > >
> > > > Q5 In the same section, in figure 2, it would be good to see for which values in the x-axis, the performance plateaus (it seems to continue to get better for the values shown).
> > > > Q6 Another comment to that section and figure, if that is possible, it would be much more useful to view a 2D grid of performance as a function of distance threshold and codebook.
> > >
> > > We agree that, in principle, identifying where reconstruction performance plateaus as a function of codebook size / cutoff would be informative. In this work, however, our objective is not to optimize reconstruction error, but to find a discrete graph representation that is good enough while remaining tractable and reliable for generation. Our current results are sufficient to make the choice of these hyperparameters. Please refer to our response to W2 for the full rationale.
> > >
> > > Due to the limited time in this cycle, we could not perform these experiments, but we believe our current experiments are sufficient to support all out claims. We are happy to include these results in final version if necessary.
> > >
> > > > Q7 Related to a point brought up in the weaknesses section, but the motivation to settle for codebook size 256 and distance threshold 8 is not at all explained.
> > >
> > > Please refer to our response to W2.
> > >
> > > > Q8 In section 5.2, sequence quality is evaluated through FID of ESM2 embeddings. There exist alternatives to FID that claim to be better (such as Feature Likelihood Divergence, PQMass) it would be interesting to see results on those as well if possible
> > >
> > > Thank you for your suggestion. We want to emphasize that the aim of Section 5.2 is not to conduct a metrics study, but to verify that AutoFold produces sequences with reasonable quality and diversity in a way that is consistent with prior work. FID is widely used in protein structure generation [3] and we naturally chose it for measuring our sequence generation quality. Coupled with our sequence diversity metrics, it already supports the main claim we need here.
> > >
> > > A careful, systematic comparison of sequence-quality metrics for proteins remains an open question for the field. Since resolving that question is outside the scope of this paper, we leave it to future work.
> > >
> > > > Q9 In table 1, the smaller Autofold seems to do better in some categories. It would be good to get an explanation for this behaviour.
> > >
> > > Thank you for your observation. These metrics should be interpreted jointly, rather than interpreting each separately. Our interpretation is that the results reflect multi-objective trade-offs and model–sampling interactions.
> > >
> > > Several quantities in Table 1 move in opposite directions by design (e.g., novelty vs. designability; diversity vs. co-designability). A smaller model can score better on one axis because it explores a slightly different region of the trade-off surface, even if it is not globally “better.” For example, a model that is more conservative can yield higher designability but lower novelty; a model that is more exploratory can do the reverse.
> > >
> > > With the same decoding scheme, larger models often produce sharper, higher-confidence distributions (Our training loss for larger models are always lower than smaller models). This typically improves co-designability but can reduce diversity/novelty. Smaller models produce a softer distribution, which can increase diversity or novelty at some cost to co-designability. So “better in some categories” is consistent with the expected size-dependent exploration–exploitation balance.
> > >
> > > This phenomenon is pretty common in this field. For example, La-Proteina [4] demonstrates very different metric profiles even if using the same model, but with different inference hyperparameters. We have added a brief clarification in the caption of Table 1 emphasizing that these results should be read as different operating points on a shared trade-off surface, rather than as independent leaderboards per metric.

---

> > > > ### Author Response · Authors · 2025-11-24
> > > > **Author response to Reviewer pLGR (cont.)**
> > > >
> > > > > Q10 Similarly, there is no comment about the big gap in designability compared to La Proteina
> > > >
> > > > Thank you for pointing this out. We agree that the designability gap relative to La-Proteina deserves an explicit comment, and we have added a discussion in Section 5.2 of the revised manuscript.
> > > >
> > > > Our key point is that **designability should be interpreted jointly with novelty**, because the two are tightly coupled. There is a well-known trade-off: the more novel a generated backbone is, the harder it is for a sequence designer such as ProteinMPNN, trained on native structures, to find a sequence that folds back to it. This mismatch can then propagate to ESMFold, lowering the measured designability even when the backbone is physically plausible (this is also relected in our results where AutoFold sometimes achieves even higher co-designability and designability). In other words, lower designability can be an expected consequence of operating in a more novel region of structure space, rather than a direct indicator of implausible backbones. Since AutoFold achieves much better novelty, it is expected that its structures are less designable when using ProteinMPNN.
> > > >
> > > > > Q11 There is also no discussion (as far as I can see) about the choice of parameters for generation, such as temperature, and how they affect these results. In table 2, the results for AutoFold are much better with a distance threshold of 9. This once again raises the question of why 8 was chosen.
> > > >
> > > > - Inference hyperparameters. Our decoding settings were originally selected to maximize co-designability, since this is the primary practical criterion for many downstream uses. As discussed in our rebuttal to the secondary-structure bias concern, we also re-selected decoding hyperparameters with the explicit goal of increasing beta-sheet content, which leads to a more balanced helix/sheet distribution. Crucially, these are inference-time knobs: practitioners can readily tune them to match their application. This is standard in the field; for example, La-Proteina reports results for the same trained model under multiple decoding settings.
> > > > - Distance cutoff. Please see our response to W2 for the general rationale of using a cutoff of 8 Å. The 9.0 Å model (AutoFold-long) in Table 2 was trained for a specific diagnostic purpose tied to Section 5.3: a current limitation of our method is scaffolding motifs whose key residues are separated by distances beyond the cutoff. If these long-range geometric relationships are not represented as edges, the conditional prompt lacks essential constraints. Increasing the cutoff partially alleviates this by incorporating longer-range contacts, which is why AutoFold-long improves those cases. We included this variant to demonstrate the directionality of the fix, and note that it comes with only a negligible sampling-time increase. In other words, 8.0 Å is our default sparse representation chosen to support the main methodological aim of the paper, while 9.0 Å is a targeted extension illustrating how to address a limitation for long-range motifs. While increasing this cutoff might improve unconditional generation performance, our experiments show that 8 Å is sufficient to reconstruct the backbone coordinates and it will come at the cost of reducing the sparsity of the graph.
> > > >
> > > > We have incorporated this discussion on the choice of these hyperparameters in Section 5.1 and 5.2 of the revised manuscript.
> > > >
> > > > [1]: Rives, Alexander, et al. Biological structure and function emerge from scaling unsupervised learning to 250 million protein sequences. PNAS 2021.
> > > > [2]: Bissantz, Caterina, Bernd Kuhn, and Martin Stahl. A medicinal chemist’s guide to molecular interactions. Journal of Medicinal Chemistry 2010.
> > > > [3]: Geffner, Tomas, et al. Proteina: Scaling Flow-based Protein Structure Generative Models. ICLR 2025.
> > > > [4]: Geffner, Tomas, et al. La-proteina: Atomistic protein generation via partially latent flow matching. arXiv preprint 2025.

---

### Official Review · Reviewer_Cd7Q · 2025-11-01

**Soundness:** 3
**Presentation:** 3
**Contribution:** 3
**Rating:** 4
**Confidence:** 5

**Summary:**

AutoFold is an autoregressive model for protein sequence and structure generation that operates by generating a sparse, discrete contact graph representation instead of continuous 3D coordinates. It uses a Vector Quantized Variational Autoencoder to encode protein backbone geometry into graph edges and then trains an autoregressive transformer to sequentially generate these graphs, incorporating sequence information as node features. This method enables co-generation and motif scaffolding at speeds over an order of magnitude faster than the leading diffusion and flow matching models while maintaining competitive design quality and diversity. The framework naturally supports zero-shot motif scaffolding and large-scale protein design, though it shows bias toward alpha-helical structures due to its graph representation choices.

**Strengths:**

- Novel architecture and use of the VQ-VAE to encode backbone structure.
- Promising scaling of the transformer decoder compared to prior work.
- Superior generation speed.
-Ability to do conditional tasks without finetuning as well as simple sequence co-conditioning coupling
- Orders of magnitude better novelty (which oddly is not emphasized)

**Weaknesses:**

- Given that AutoFold is autoregressive and it randomly generate 500 proteins for AutoFold models without explicit length controls, what is the distribution of the lengths used in the benchark for Table 1? Shorter proteins are typically much easier to co-design so the performance many be better or worse when normalized for length.
- 90% helix and nearly no beta sheet is a major limitation. Especially when ESMFold designability has a strong bias towards helicity.
- Not the most fair motif benchmark. La Proteina and Protpardelle do all atom motif scaffolding whereas the backbone only motif tasks used would be a better comparison to prior backbone methods benchmarked in Proteina [1] Table 5 albeit AutoFold has a sequence component. Also the motif tasks by design have a length requirement.
- Missing comparison to MultiFlow https://arxiv.org/abs/2402.04997

**Questions:**

- Can a length based comparison be made by some amount of oversampling filtering, and binning to evaluate the first K proteins that are close to the length binds of the prior diffusion and flow models?
- The novelty is orders of magnitude better than every prior method. Any reason as to why? Given the samples are 90% helix it seems odd to see such strong AFDB novelty which is known to be helix saturated.
- How are the length requirements respected for the motif benchmarking

---

> ### Author Response · Authors · 2025-11-24
> **Author response to Reviewer Cd7Q**
>
> We thank you for your thoughtful and constructive feedback. Please find our response to each of your points below.
>
> > W1. Given that AutoFold is autoregressive and it randomly generate 500 proteins for AutoFold models without explicit length controls, what is the distribution of the lengths used in the benchark for Table 1? Shorter proteins are typically much easier to co-design so the performance many be better or worse when normalized for length.
>
> Please refer to our general response "Concern on potentially unfair comparison due to length-unconstrained generation".
>
> > W2. 90% helix and nearly no beta sheet is a major limitation. Especially when ESMFold designability has a strong bias towards helicity.
>
> Please refer to our general response "Concern on strong alpha-helix overrepresentation".
>
> > W3. Not the most fair motif benchmark. La Proteina and Protpardelle do all atom motif scaffolding whereas the backbone only motif tasks used would be a better comparison to prior backbone methods benchmarked in Proteina [1] Table 5 albeit AutoFold has a sequence component. Also the motif tasks by design have a length requirement.
>
> We thank the reviewer for raising this benchmarking concern. We agree that comparing across motif-scaffolding setups is requires subtlety, especially when methods differ in resolution and conditioning. However, we respectfully believe the benchmark we use is the most appropriate for the current scope of AutoFold, for the following reasons:
>
> - **AutoFold is a sequence–structure co-design model, not a structure-only generator.** Our motif scaffolding outputs both backbones and sequences, and we evaluate using co-designability and related metrics that are directly aligned with La-Proteina’s intended use case (up to the side-chain resolution). In this sense, our task formulation is closer to La-Proteina/Protpardelle than to Proteina, which is structure-only and does not jointly model sequences. Using a structure-only benchmark would therefore understate the capabilities and the objective of our model.
> - **All-atom extension is conceptually straightforward.** AutoFold can be extended to all-atom motif scaffolding by augmenting the VQ-VAE decoder with a side-chain reconstruction head conditioned on the amino-acid sequence, analogous to the structural-token decoder used in ESM3 (see Appendix A.1.7 in [1]). Implementing this in a robust, competitive way requires substantial engineering and computing, and the relevant ESM3 components are not publicly released, so we leave a full all-atom instantiation to future work. Importantly, this does not change the core generative mechanism we propose.
> - **Structure-only baselines behave qualitatively differently under conditioning.** Proteina-style models are trained to generate backbone structure alone and rely on separate inverse folding modules for sequence design, typically ProteinMPNN. Therefore, the design results could also be affected by the quality of the inverse folding model. In contrast, our model does not rely on such models. As a result, mixing structure-only conditional benchmarks with co-design conditional benchmarks makes it harder to interpret differences and can lead to misleading conclusions.
>
> > W4 Missing comparison to MultiFlow https://arxiv.org/abs/2402.04997
>
> Due to the limited time in this cycle, we haven't finished the evaluation for MultiFlow. As this method is a diffusion model, we believe its results would not affect our main claims in the paper. We will update our response and add its results to our Table 1 once we finish the experiments.

---

> > ### Author Response · Authors · 2025-11-24
> > **Author response to Reviewer Cd7Q (cont.)**
> >
> > > Q1 Can a length based comparison be made by some amount of oversampling filtering, and binning to evaluate the first K proteins that are close to the length binds of the prior diffusion and flow models?
> >
> > Please refer to our general response "Concern on potentially unfair comparison due to length-unconstrained generation".
> >
> > > Q2 The novelty is orders of magnitude better than every prior method. Any reason as to why? Given the samples are 90% helix it seems odd to see such strong AFDB novelty, which is known to be helix saturated.
> >
> > We first want to clarify an important confounder highlighted in our general response "Concern on strong alpha-helix overrepresentation", extremely helix-rich samples can inflate novelty by producing long helices. Therefore, we trained AutoFold-m* on the secondary-structure-filtered AFDB subset used by La-Proteina, yielding a helix/beta balance comparable to La-Proteina. Under this matched setting, AutoFold-m* achieves unique co-designability on par with La-Proteina, while still retaining strong novelty (AutoFold-m*: 0.48 vs. La-Proteina: 0.77; lower is better in our novelty metric).
> >
> > While we do not know the full reason for this high novelty, we hypothesize that this could stem from the separation of responsibilities in our pipeline:
> > - VQ-VAE provides a strong local structural prior
> > - The autoregressive graph model explores global topologies in a discrete, combinatorial space with explicit long-range dependencies.
> >
> > This combination allows AutoFold to move into novel structure space while staying within a learned manifold of locally valid backbones, leading to high novelty without catastrophic loss of foldability. We have included a discussion in Section 5.2 of our revised manuscript.
> >
> > > Q3 How are the length requirements respected for the motif benchmarking
> >
> > At inference time, we restrict the allowable index tokens to a prescribed range (e.g. in $\{1,...,L_{\max}\}$). This guarantees that the model cannot generate indices beyond $L_{\max}$, and thus cannot produce proteins longer than the benchmark length cap. Conversely, minimum-length requirements can be enforced by adding complementary constraints (e.g., requiring that all indices up to $L_{\min}$ be generated before EOS). Because these constraints operate directly on the index-token stream, they are integrated cleanly with our syntactic/semantic validity constraints for graph decoding. We have clarified this decoding process in Appendix C.4 of the revised manuscript.

---

### Official Review · Reviewer_dGv6 · 2025-11-06

**Soundness:** 2
**Presentation:** 2
**Contribution:** 3
**Rating:** 4
**Confidence:** 5

**Summary:**

This paper presents AutoFold, a novel two-stage method for de novo protein structure and sequence co-generation. The core idea is to move from continuous coordinate generation (common in diffusion models) to a discrete, autoregressive process. In the first stage, a VQ-VAE is trained to learn a discrete "vocabulary" of geometric features, effectively tokenizing a protein's 3D structure into a sparse latent contact graph. In the second stage, an autoregressive model (based on the AutoGraph framework) is trained to generate these graphs, with amino acid sequences treated as node attributes. The authors claim this approach achieves strong performance on par with state-of-the-art (SOTA) models for unconditional generation and motif scaffolding, while being over an order of magnitude faster to sample.

**Strengths:**

1. The primary strength of this paper is its novel approach. Reframing the protein generation problem as an autoregressive task on a discretized, latent contact graph is an interesting idea that moves away from the dominant diffusion/flow-matching paradigm.
2. The model demonstrates impressive sampling speed, reportedly orders of magnitude faster than most baselines. This speed, combined with competitive performance on co-designability and novelty, makes it a promising direction for high-throughput design.
3. The two-stage approach (VQ-VAE for representation, AutoGraph for generation) is a clever way to simplify the generation problem, making it tractable for a powerful sequence-based model. The ability to perform zero-shot motif scaffolding is also an advantage.

**Weaknesses:**

Despite its strengths, the paper suffers from several significant weaknesses, primarily concerning methodological clarity and the experimental setup.
1. The paper is not self-contained, particularly regarding the core autoregressive mechanism in Section 4.2. The authors state it is built based on an existing framework AutoGraph and refer readers to the original paper. However, this graph generation step is central to the paper's contribution, and its application to the protein domain is non-trivial. The paper would be much stronger if it expanded this section with more detail (moving specifics to an appendix if needed) to explain how the graph is "flattened", how the stochastic ordering is generated, and how validity is ensured.
2. The model is termed "autoregressive," but it does not generate the protein in its natural sequence order. Instead, it uses a stochastic graph traversal order, which appears to treat the protein as a permutation-invariant contact graph. This is a strong, and perhaps incorrect, inductive bias, as proteins are fundamentally not permutation-invariant; their properties are tied to the N-to-C terminal sequence. This choice needs to be explicitly justified and discussed.

Experimental Concerns:
1. The VQ-VAE's reconstruction RMSD of >0.6A (per Fig. 2, ~0.8A for the chosen 256 codebook size) seems significantly higher than the fidelity of other models (e.g., La-Proteina's VAE at ~0.12A). While "sub-angstrom" is good for folding, this level of reconstruction error in the VAE stage may be a weak link.
2. The primary comparison in Table 1 is potentially unfair. AutoFold is length-unconstrained, while the baselines are length-conditioned. It is known that generating longer proteins is more difficult. Without seeing the length distribution of AutoFold's generated samples, it is impossible to know if it is "hacking" the metrics by primarily generating shorter, easier proteins. The fact that all sample visualizations appear short is concerning. The authors must provide this length distribution for a fair comparison.
3. Several generative models, such as DPLM-2 and MultiFlow, are absent from the table.
4. The "unique co-designability" metric is confusing. Based on the definition, it is equivalent to the number of designable clusters which should be an integer, but the values in Table 1 are not integers. This should be clarified.
5. The model's strong overrepresentation of alpha-helices (~90%) is a major weakness, which the authors acknowledge. This suggests the model is not capturing the full diversity of protein structures, and it's unclear if this is due to the methodology (e.g., the VQ-VAE representation) or the sampling scheme.
6. While the model is clearly fast, the claim of "over an order of magnitude" faster sampling seems to apply only to older diffusion models (like RFDiffusion), not the SOTA flow-matching model La-Proteina. The speedup over La-Proteina (per Fig. 3) is closer to 3x, which is still good but not an order of magnitude.

**Questions:**

To address the weaknesses above, I have the following questions:

1. **On Graph Generation** (Sec 4.2): Could you please elaborate on the graph flattening process?
- What is the specific meaning of the '<' and '>' tokens in the sequence in Figure 1?
- Why are node indices (e.g., node '3') generated multiple times in the example sequence?
- How exactly is the stochastic ordering generated during training?
- During inference, how do you ensure the generated token sequence is syntactically valid and decodes to a valid graph?
- How is the graph sparsity (i.e., the total number of nodes and edges) controlled or learned during generation?
2. **On Model Choice**: Why use a stochastic graph ordering, which implies permutation invariance, instead of the protein's canonical N-to-C sequence order? Given that this choice seems to discard crucial biological information, what is the authors' interpretation of its impact? What advantages does this specific "autoregressive" formulation have over any-order generation models like diffusion?
3. **On Experimental Results**:
- Have you investigated the source of the strong alpha-helical bias? Is it a consequence of the low-temperature sampling, or is it an inherent bias of the VQ-VAE representation, which may find it easier to tokenize helices?
- Your model shows higher co-designability than PMPNN-8 designability, a result hardly seen in previous models. What is your interpretation of this? Is it possible the model is "hacking" this metric by overrepresenting specific residue types that are "easy" to co-design (which might also explain the alpha-helix bias)?
- The model achieves both good co-designability and high novelty. This is an excellent result. Do you have an interpretation for why this method is particularly good at finding novel, valid designs?

In general, I think this paper presents a good, novel idea with promising results. However, the paper is currently held back by a significant lack of methodological detail and a few concerning issues in the experimental setup (length bias, alpha-helix bias, metric clarity). I'm happy to discuss more during the rebuttal process and increase my score if my concerns are addressed.

---

> ### Author Response · Authors · 2025-11-24
> **Author response to Reviewer dGv6**
>
> We thank you for your thoughtful and constructive feedback. Please find our response to each of your points below.
>
> > Paper not self-contained on autoregressive graph generation (Sec. 4.2)
>
> While we cannot reproduce the full AutoGraph paper within the main text, we believe the current manuscript already contains the essential ingredients in Section 4.2 and Appendix C.4.
>
> Concretely, for each graph $G = (V, E)$, AutoGraph samples from this graph a random path sequence with restarts and neighborhood information $w = (w_1, ..., w_n)$, termed Segmented Eulerian Neighborhood Trail (SENT), where $w_k = (v_k, A_k)$ with a node (index) $v_k \in V$ and $A_k \subseteq V$ is the set of all previously visited nodes that are neighbors of $v_k$. This path (with restarts) visits each node and edge exactly once, making it a concise and lossless sequential representation of the graph. **More importantly, each prefix of this sequence generates an induced subgraph of $G$, making it possible for substructure-conditioned generation**.
>
> To model SENT with a language model, we convert $w$ into a sequence of tokens by expanding all neighborhood sets and inserting special delimiter tokens. These tokens include symbols such as ‘/’ to indicate a breakage between segments, and ‘<’ and ‘>’ to mark the start and end of a neighborhood set.
>
> During inference, we use constrained decoding (Sec. 4.3, App. D) so that the generated token stream is syntactically and semantically valid and always decodes to a valid graph.
>
> **We have updated Section 4.2 and the caption of Figure 1 to incorporate this clarification in the revision.** Note that the key advantage of this flattening/generation scheme is that it allows **one single model to support both unconditional generation and motif scaffolding** without retraining, **unlike diffusion/flow matching methods, such as La Proteina**, that typically require costly task-specific adaptation.
>
> If you need any further specific clarifications, we would be happy to include them.
>
> > W2. Stochastic ordering and permutation invariance
>
> We agree that proteins are not permutation-invariant objects. Our model **does not discard sequence order** and does not treat proteins as permutation-invariant graphs. As stated at the end of Section 4.2, the contact graph is an **ordered graph**: nodes carry their canonical residue indices, and these indices are the node IDs emitted in the SENT sequence. Therefore, the original N→C order is always recoverable from the generated graph/sequence, and the model is trained on this ordered representation.
>
> The stochastic traversal order is used only to define an autoregressive factorization over graphs, not to impose invariance. This is to enable substructure-conditioned generation (i.e. motif scaffolding). In this way, a sufficient number of substructures can be sampled as prefixes of the token sequences during training so that our model can be used for motif scaffolding out of the box.
>
> We have added a clarification in Section 4.2 in the revised manuscript.
>
> > WEC1. The VQ-VAE's reconstruction RMSD of >0.6A (per Fig. 2, ~0.8A for the chosen 256 codebook size) seems significantly higher than the fidelity of other models (e.g., La-Proteina's VAE at ~0.12A). While "sub-angstrom" is good for folding, this level of reconstruction error in the VAE stage may be a weak link.
>
> The reconstruction tasks between our method and La-proteina are not directly comparable:
> - La-Proteina learns a latent over non-Cα coordinates given the full atom coordinates and the sequence, and reconstructs them **given the Cα coordinates and the latent variables**. Non-Cα atoms are close to Cα geometry and the latents only need to encode local geometry, making the reconstruction substantially easier.
> - AutoFold learns a _discrete latent representation_ of the full backbone and reconstructs backbone coordinates **from latents alone**, i.e., without conditioning on the Cα coordinates. This is a harder task because the latent must encode the global backbone geometry directly.
>
> To calibrate difficulty, we point to ESM3, which also learns discrete backbone latents; their reconstruction error is comparable to ours, even with a 4 times larger codebook (See their Figure S3.B). We have added a clearer “task-difficulty” comparison in Section 5.1 in the revised manuscript to avoid misleading cross-method RMSD comparisons.

---

> > ### Author Response · Authors · 2025-11-24
> > **Author response to Reviewer dGv6 (cont.)**
> >
> > > WEC2. Potentially unfair comparison due to length-unconstrained generation
> >
> > We agree that length effects can bias unconditional metrics. We therefore provide the length distribution of generated samples in Appendix E.4 in the revised manuscript.
> >
> > More broadly, we want to stress the intent of this work: as the first autoregressive contact-graph model for proteins, our goal is not to claim unconditional SOTA, but to show that (i) protein generation is feasible via sparse contact-graph generation, and (ii) a single AR model can handle both unconditional and substructure-conditional generation while remaining in a reasonable designability regime.
> >
> > > WEC3. Several generative models, such as DPLM-2 and , are absent from the table.
> >
> > Due to limited time in this cycle, we haven't finished the evaluation for MultiFlow and DPLM-2. As both methods are diffusion models, we believe their results would not affect our main claims in the paper. We have added DPLM-2 in our related work (Multiflow was already discussed), and will update our response and add the results for both methods to our Table 1 once we finish the experiments.
> >
> > > WEC4. Unique co-designability definition is confusing
> >
> > We apologize for the imprecision. Unique co-designability is not an integer count, but a ratio: the number of Foldseek clusters among co-designable samples divided by the total number of generated samples. We have revised the definition in the manuscript.
> >
> > > WEC5. Strong alpha-helix overrepresentation
> >
> > Please refer to our general response "Concern on strong alpha-helix overrepresentation".
> >
> > > WEC6. Speedup claim vs La-Proteina
> >
> > Thank you for flagging this. We agree that “over an order of magnitude” is accurate only relative to diffusion baselines; the speedup over La-Proteina is closer to ~3x. We have updated the abstract and introduction accordingly in the revised manuscript.
> >
> > > Q1 On Graph Generation (Sec 4.2): Could you please elaborate on the graph flattening process?
> >
> > Please see our response to W1.
> >
> > > Q1.1 What is the specific meaning of the '<' and '>' tokens in the sequence in Figure 1?
> >
> > They mark the start/end of the neighborhood set $A_k$, i.e., the previously visited neighbors associated with the currently visited node.
> >
> > > Q1.2 Why are node indices (e.g., node '3') generated multiple times in the example sequence?
> >
> > A node index appears (i) when first visited, with its amino acid label, and (ii) later as an element of a neighborhood set for another visited node. Repetition does not mean revisiting; it encodes edges succinctly.
> >
> > In our figure, the first '3' with a node label 'T' indicates that the trail first visits this node, representing the residue 'T' in the amino acids. The second and third '3' mean that it is the neighbor of the currently visited node 5 (labeled as 'V') and node 2 (labeled as 'K').
> >
> > We have updated the caption and Section 4.2 in our revised manuscript.
> >
> > > Q1.3 How exactly is the stochastic ordering generated during training?
> >
> > It is generated by random path sampling on the graph. Different samples correspond to different valid trails/orderings, used during training to define multiple autoregressive factorizations. For example, a depth-first search order can be such a sample.
> >
> > > Q1.4 During inference, how do you ensure the generated token sequence is syntactically valid and decodes to a valid graph?
> >
> > As described in the first paragraph of Section 4.3 and Appendix D, we use constrained decoding to make sure the generated token sequence is semantically correct.
> >
> > > Q1.5 How is the graph sparsity (i.e., the total number of nodes and edges) controlled or learned during generation?
> >
> > The graph sparsity is controlled by the distance cutoff, which is a hyperparameter. We chose 8 Å as it is a common choice for the contact map [1] primarily because a majority of local intermolecular interactions manifest within this range [2]. It yields an average degree $\approx 8$ in our dataset.
> >
> > > Q2.1 Why use a stochastic graph ordering, which implies permutation invariance, instead of the protein's canonical N-to-C sequence order?
> >
> > As discussed in W2, the stochastic ordering is a generation order, not a claim of permutation invariance. The model still operates on ordered graphs with residue indices, preserving N→C information.
> >
> > > Q2.2 Given that this choice seems to discard crucial biological information, what is the authors' interpretation of its impact?
> >
> > We don't think this question is relevant anymore.

---

> > > ### Author Response · Authors · 2025-11-24
> > > **Author response to Reviewer dGv6 (cont.)**
> > >
> > > >  Q2.3 What advantages does this specific "autoregressive" formulation have over any-order generation models like diffusion?
> > >
> > > We want to clarify that our primary contribution is not the choice of an "autoregressive" generator itself, but rather the introduction of a compact, sparse, discrete contact-graph representation of proteins that is suitable for generative modeling. We adopt a state-of-the-art AR graph generator as a first proof-of-concept on this representation, but we fully agree that other generative paradigms, e.g., diffusion or flow-matching models for discrete graphs [3], could also be built on top of the same representation.
> > >
> > > That said, pairing this representation with an AR formulation brings concrete advantages in our setting:
> > >
> > > - **Efficiency and scalability on sparse structure graphs.** Working on sparse contact graphs is substantially cheaper than operating on dense residue-pair representations, typically used in diffusion/flow matching models. AR generation naturally exploits this sparsity, enabling fast generation.
> > > - **A single model for both unconditional and substructure-conditional generation.** AutoFold trains one autoregressive model that supports unconditional sampling and motif scaffolding through the same generative mechanism, without task-specific retraining.
> > > - **Token-based modeling that aligns with multimodal LLM paradigms.** The discrete graph tokens provide a natural interface to language-modeling machinery. This opens the door to multimodal protein LLMs that can jointly reason over sequence and (sub)structure in a unified token space, and to flexible conditional generation via text prompting.
> > >
> > >
> > > > Q3.1 Have you investigated the source of the strong alpha-helical bias? Is it a consequence of the low-temperature sampling, or is it an inherent bias of the VQ-VAE representation, which may find it easier to tokenize helices?
> > >
> > > Please refer to our general response "Concern on strong alpha-helix overrepresentation".
> > >
> > > > Q3.2 Your model shows higher co-designability than PMPNN-8 designability, a result hardly seen in previous models. What is your interpretation of this? Is it possible the model is "hacking" this metric by overrepresenting specific residue types that are "easy" to co-design (which might also explain the alpha-helix bias)?
> > >
> > > Our model is not hacking this metric. In fact, there is a trade-off between designability and novelty: highly novel backbones are, by construction, more challenging for a sequence-design model such as ProteinMPNN, which is trained on native protein structures. This difficulty can then propagate to ESMFold, leading to lower designability even if the generated backbone is physically plausible. We have included a discussion in Section 5.2 of our revised manuscript.
> > >
> > > > Q3.3 The model achieves both good co-designability and high novelty. This is an excellent result. Do you have an interpretation for why this method is particularly good at finding novel, valid designs?
> > >
> > > We first want to clarify an important confounder highlighted in our general response "Concern on strong alpha-helix overrepresentation": extremely helix-rich samples can inflate novelty by producing long helices. Therefore, we trained AutoFold-m* on the secondary-structure-filtered AFDB subset used by La-Proteina, yielding a helix/beta balance comparable to La-Proteina. Under this matched setting, AutoFold-m* achieves unique co-designability on par with La-Proteina, while still retaining strong novelty (AutoFold-m*: 0.48 vs. La-Proteina: 0.77; lower is better in our novelty metric).
> > >
> > > While we do not know the full reason of this high novelty, we hypothesize that this could stem from the separation of responsibilities in our pipeline:
> > > - VQ-VAE provides a strong local structural prior
> > > - The autoregressive graph model explores global topologies in a discrete, combinatorial space with explicit long-range dependencies.
> > >
> > > This combination allows AutoFold to move into novel structure space while staying within a learned manifold of locally valid backbones, leading to high novelty without catastrophic loss of foldability. We have included a discussion in Section 5.2 of the revised manuscript.
> > >
> > > [1]: Rives, Alexander, et al. Biological structure and function emerge from scaling unsupervised learning to 250 million protein sequences. PNAS 2021.
> > >
> > > [2]: Bissantz, Caterina, Bernd Kuhn, and Martin Stahl. A medicinal chemist’s guide to molecular interactions. Journal of medicinal chemistry 2010.
> > >
> > > [3]: Vignac, Clément, et al. DiGress: Discrete Denoising diffusion for graph generation. ICLR 2023.

---

> > > > ### Comment · Reviewer_dGv6 · 2025-11-26
> > > >
> > > > Thanks for the authors' detailed response. I do see most of my concerns are addressed in the reseponse, especially the interpretation of alpha-helices overrepresentation and unfair compairson due to unbalanced length distribution. As the paper shown, the length distribution is ineed biased towards short proteins. Nevertheless, this should be seen as an inherent property of autoregressive model and it is hard to conclude whether this is good or not in practice. Overall, I think the results and analysis are interesting. I'll update my scores to 6.

---

> > > > > ### Author Response · Authors · 2025-11-27
> > > > >
> > > > > Thank you for your response and support! We are happy to continue the discussion if you have any other questions. We will add the results for MultiFlow and DPLM-2 once we have them.

---

### Official Review · Reviewer_NzJQ · 2025-11-06

**Soundness:** 3
**Presentation:** 3
**Contribution:** 2
**Rating:** 4
**Confidence:** 5

**Summary:**

This work presents a two-stage de novo protein generation pipeline that first trains a VQVAE on protein frames, and then uses the resulting codebook to autoregressively generate 3D protein backbone structures.

**Strengths:**

1. The codebooks in VQ-VAEs are usually unstable/erratic to train but AutoFold's reconstruction low recon-RMSD of 0.7 angstrom is noteworthy.

2. The generation speed of proteins with AutoFold is fast compared to existing SOTA methods, which could be useful in high-throughput design pipelines.

**Weaknesses:**

1. The baselines all operate at an all-atom resolution, generating the sidechain atoms/identities as well. AutoFold only operates on the backbone resolution (CA, C, N, O), which makes the modeling task much easier. The community has moved on from backbone-only generation for quite some time now. Could the authors comment on efforts / potential extensions to convert it into an all-atom model?

2. Usually, there is a huge trade-off between Designability (% foldable by ESMFold) and Novelty metrics. If a protein design method generates a very unique backbone, it means P-MPNN might have some difficulty producing a sequence that's likely to fold into it. This error further propagates to ESMFold that may predict a 3D structure that has low self-consistency with the generated backbone. This means it's less likely to meet the designability threshold of scRMSD<2 angstroms despite being very novel. I think this might be happening here: for the best-performing AutoFold model with a good balance of scores, the all-designability metric (50.4) is rather low compared to current SOTA, La Proteina (72%), but has higher novelty (0.718 vs 0.571 respectively). It's difficult to offer an apples-to-apples comparison between models when this is the case.

3. The community is more concerned with the Designability and Novelty metrics being high than inference speed. La Proteina-like methods are already pretty fast for large sequence lengths, and AutoFold is not that much faster, especially for proteins of size ~800. Also, given the autoregressive nature of AutoFold, I'm skeptical if AR-style generation is really superior than diffusion/flow-style generation.

**Questions:**

1. I'm curious to see the % of secondary structural elements like alpha helices and beta sheets. The issue with autoregressive protein design models (eg: FoldingDiff [1]) is that they have this tendency to generate multiple helices because they are more well-represented than loops and beta sheets. Perhaps a figure like Figure 3 in [FoldFlow2's paper](https://arxiv.org/abs/2405.20313) would be useful in showing us AutoFold's local structural diversity as well.

2. All of these protein design methods (including AutoFold) are trained on different subsets of the PDB/AFDB, which makes the comparison problem even harder since some methods are more capable than others in designing novel structures.

3. The authors mention they have a sequence length cut-off of 256 residues. **To convince me further, I'd be interested in seeing how this method scales to larger proteins > 300 residues** (if time permits). Authors mention their method can be scaled up to such lengths, but there's no evidence of this. Furthermore, La Proteina trains an all-atom model upto 896 residues (understandable, given that it's NVIDIA and they might have the compute resources) and still performs really well. Authors provide Figure 3 detailing the inference speed for increasing lengths. I'd want to also see how designability and other metrics differ as AutoFold is scaled up to increasing lengths. A figure of seq-len vs average designability (vs La Proteina) would be very convincing.

[1] Protein structure generation via folding diffusion. Wu et al. (2022)

---

> ### Author Response · Authors · 2025-11-24
> **Author response to Reviewer NzJQ**
>
> We thank you for your thoughtful and constructive feedback. Please find our response to each of your points below.
>
> > W1. Backbone vs all-atom generation: Could the authors comment on efforts / potential extensions to convert it into an all-atom model?
>
> We agree that all-atom models are important for end-to-end design, and we view our current backbone-only formulation as a first step toward all-atom generation.
> Backbone-level representations remain central in many practical workflows, where side chains could be added using specialized decoders or packing models. Our contribution is orthogonal to this choice of resolution: the autoregressive contact-graph representation and training procedure apply equally well if one can learn a tokenization that encodes all-atom configurations instead of backbone-only coordinates.
>
> In particular, **AutoFold can be extended in a conceptually straightforward way to the all-atom setting by pairing VQ-VAE decoder with a side-chain reconstruction head trained jointly or in a subsequent stage by additionally taking the amino acid sequence as input, similar in spirit to ESM3's VQ-VAE decoder, where a decoder maps structural tokens and sequences to full-atom coordinates (More details can be found in Appendix A.1.7 in [1])**. Our loss function can naturally incorporate a full-atom reconstruction term instead of the backbone atom reconstruction term. This would only result in a different VQ-VAE decoder that takes both the graph and the sequence as input, and reconstructs the full atom coordinates. We see this as a promising direction for follow-up work and believe that the main methodological insights of AutoFold, contact-graph tokenization, autoregressive generation, and unified unconditional/motif-conditional modeling may carry over directly to the all-atom regime.
>
>
> > W2. designability–novelty trade-off and comparison to La Proteina
>
> We fully agree that there is a trade-off between designability and novelty: highly novel backbones are, by construction, more challenging for a sequence-design model such as ProteinMPNN, which is trained on native protein structures. This difficulty can then propagate to ESMFold, leading to lower designability even if the generated backbone is physically plausible.
>
> We also want to clarify an important confounder highlighted in our general response "Concern on strong alpha-helix overrepresentation": extremely helix-rich samples can inflate novelty by producing long helices. Therefore, we trained AutoFold-m* on the secondary-structure-filtered AFDB subset used by La-Proteina, yielding a helix/beta balance comparable to La-Proteina. Under this matched setting, AutoFold-m* achieves unique co-designability on par with La-Proteina, while still retaining strong novelty (AutoFold-m*: 0.48 vs. La-Proteina: 0.77; lower is better in our novelty metric). As a consequence, we see AutoFold as exploring a more novel regime without sacrificing foldability.
>
> We therefore view the two approaches as complementary rather than strictly competing. Diffusion/flow matching models such as La-Proteina set an excellent bar for designability in more conservative regions of fold space, while AutoFold shows that an autoregressive contact-graph model can maintain reasonable designability while pushing further into structurally diverse and novel backbones. The corresponding reduction in ProteinMPNN-designability is thus consistent with the expected novelty–designability trade-off.
>
> Finally, beyond unconditional metrics, AutoFold offers a practical capability that is central to our contribution: **a single trained model supports both unconditional generation and motif scaffolding**. With the autoregressive graph formulation, conditional generation is realized by flattening the desired substructure into a prompt and continuing generation, without retraining or fine-tuning. In contrast, La-Proteina and other diffusion/flow matching approaches typically require expensive task-specific retraining or adaptation. We see this unified modeling as a key advantage of our method.

---

> > ### Author Response · Authors · 2025-11-24
> > **Author response to Reviewer NzJQ (cont.)**
> >
> > > W3. The community is more concerned with the Designability and Novelty metrics being high than inference speed. La Proteina-like methods are already pretty fast for large sequence lengths, and AutoFold is not that much faster, especially for proteins of size ~800. Also, given the autoregressive nature of AutoFold, I'm skeptical if AR-style generation is really superior than diffusion/flow-style generation.
> >
> > We agree that, ultimately, practitioners care most about the quality of designed proteins, particularly diversity and novelty, rather than raw inference speed. Our speed measurements are intended to show that generating sparse contact graphs is fast and effective in contrast to the dense pair representations widely used in diffusion/flow matching models. We also note that if the method is sufficiently fast, designability is less important, as one could always generate more samples to compensate for the reduced designability.
> >
> > We do not claim that autoregressive generation is superior to diffusion/flow-style generation. Instead, our goal is to provide a first proof-of-concept that an autoregressive contact-graph model can jointly handle unconditional and substructure-conditional generation with a single model, while remaining in a reasonable designability regime. We see AR and diffusion/flow methods as complementary modeling paradigms: diffusion excels at directly modeling continuous structural manifolds, whereas AR models offer a novel perspective on protein structure modeling, and a natural way to reuse a single model across both unconditional and conditional designs. Our results show that such an AR approach is viable and promising enough to warrant further exploration.
> >
> > > Q1: I'm curious to see the % of secondary structural elements like alpha helices and beta sheets. The issue with autoregressive protein design models (eg: FoldingDiff \[1\]) is that they have this tendency to generate multiple helices because they are more well-represented than loops and beta sheets. Perhaps a figure like Figure 3 in  [FoldFlow2's paper](https://arxiv.org/abs/2405.20313)  would be useful in showing us AutoFold's local structural diversity as well.
> >
> > Thank you for your suggestion. Please refer to our general response "Concern on strong alpha-helix overrepresentation"
> >
> > > Q2: All of these protein design methods (including AutoFold) are trained on different subsets of the PDB/AFDB, which makes the comparison problem even harder since some methods are more capable than others in designing novel structures.
> >
> > We agree that the use of different training subsets across methods is a general challenge in the field and complicates strict head-to-head comparisons. In this work, we followed Genie2 and Proteina's data, but a perfectly aligned training corpus across all baselines is currently not available. We view this as a community-wide issue rather than one specific to AutoFold.
> >
> > That said, we share the reviewer’s concern and consider more standardized benchmarks with shared training sets and evaluation pipelines across methods to be an important direction for the field. Our contribution is orthogonal to the specific dataset choice, and we expect AutoFold’s relative behavior to carry over to such standardized settings once they become available.
> >
> > In this rebuttal, we also trained AutoFold-m* on the secondary-structure-filtered AFDB subset used by La-Proteina for a fairer comparison.
> >
> > > Q3: The authors mention they have a sequence length cut-off of 256 residues.  [...] A figure of seq-len vs average designability (vs La Proteina) would be very convincing.
> >
> > The current experiments restrict training to sequences up to 256 residues to keep training costs manageable for this first study of autoregressive contact graph generation. Architecturally, AutoFold scales well, and our inference speed measurements (Figure 3) already explore lengths well beyond the training cutoff, indicating that the method is computationally capable of operating in that regime.
> >
> > We agree that a systematic analysis of designability and related metrics as a function of sequence length, particularly beyond 300 residues, and in comparison to models such as La Proteina, would be more informative. This would involve not only generating longer backbones but also carefully reconsidering training data, model capacity, and computational resources. We view this as an important but substantial extension. In this paper, we deliberately concentrate on **academic-scale datasets** used by previous work to clearly establish the feasibility and properties of our autoregressive contact-graph approach.
> >
> > [1]: Hayes, Thomas, et al. Simulating 500 million years of evolution with a language model. Science 2025.

---

### Official Review · Reviewer_jUAk · 2025-11-06

**Soundness:** 3
**Presentation:** 3
**Contribution:** 3
**Rating:** 8
**Confidence:** 3

**Summary:**

This paper introduces AutoFold, a novel autoregressive generative model for de novo protein design that addresses the computational bottlenecks of existing diffusion and flow-matching approaches. By representing protein backbones as sparse, discrete contact graphs learned via a VQ-VAE (achieving ~0.8 Å RMSD reconstruction fidelity), and then generating these graphs autoregressively using a transformer-based framework adapted from AutoGraph, the method enables ultra-fast sampling (over an order of magnitude faster than baselines like RFDiffusion). It supports unconditional co-generation of sequence and structure, as well as zero-shot motif scaffolding. Evaluations on AFDB-derived datasets show competitive co-designability (~72%), diversity, and novelty compared to state-of-the-art models like La-Proteina, though with a noted bias toward alpha-helical structures. The authors plan to release code, models, and protein graph datasets.

**Strengths:**

1. Efficiency Gains: The reported speedups (e.g., 100x faster than RFDiffusion on H100 GPUs) are compelling for high-throughput applications. By flattening graphs into sequences, the model scales well with transformer architectures, and the use of constrained decoding ensures valid outputs.]
2. Strong Results: On unconditional generation, AutoFold-m achieves higher unique co-designability (46.4%) and better sequence FID (~9.07) than baselines, with good novelty against PDB/AFDB. For motif scaffolding, it solves 18/20 tasks in the benchmark, outperforming La-Proteina on flexible-length tasks (102 vs. 79 unique successes)

**Weaknesses:**

The strong preference for alpha-helices (~90% content) limits diversity, potentially due to the VQ-VAE's codebook favoring frequent local patterns. While acknowledged, more analysis (e.g., codebook utilization histograms or β-sheet-specific reconstructions) could strengthen the discussion. Comparisons to curated datasets (as in La-Proteina) might reveal if this is data- or model-induced.

The authors investigate the impact of graph sparsity by fixing the codebook size to 256 and varying the distance threshold. A fuller exploration of threshold impacts on generation complexity and speed would be useful.

**Questions:**

The unconditional generation results show a strong bias toward alpha-helical structures (~90% helix content). What steps could mitigate this for more diverse topologies? Fuller exploration on what's the root cause of the strong preference would be interesting.

---

> ### Author Response · Authors · 2025-11-24
> **Author response to Reviewer jUAk**
>
> We thank you for your thoughtful and constructive feedback. Please find our response to each of your points below.
>
> > The strong preference for alpha-helices (~90% content) limits diversity, potentially due to the VQ-VAE's codebook favoring frequent local patterns. While acknowledged, more analysis (e.g., codebook utilization histograms or β-sheet-specific reconstructions) could strengthen the discussion. Comparisons to curated datasets (as in La-Proteina) might reveal if this is data- or model-induced.
>
> Please refer to our general response "Concern on strong alpha-helix overrepresentation".
>
> > The authors investigate the impact of graph sparsity by fixing the codebook size to 256 and varying the distance threshold. A fuller exploration of threshold impacts on generation complexity and speed would be useful.
>
> We limited the exploration for practical reasons: a full grid over thresholds on generation quality and complexity would require substantial computing beyond infrastructure capacities.
>
> Our choices of these hyperparameters were based on the following considerations:
>
> - Our primary goal is to test whether protein structures admit a _usefully sparse_ contact-graph representation that can be generated through graph generation models (in our work, an autoregressive model), rather than directly modeling continuous 3D coordinates. A cutoff of 8.0 Å is a common choice for the contact map [1] primarily because a majority of local intermolecular interactions manifest within this range [2].
> - We fix the codebook size at 256 to maintain a good balance between reconstruction quality and generation difficulty. While increasing the vocab size improves the reconstruction quality, it also makes the generative task harder: for example, for the motif scaffolding task, the same motif might be encoded into a graph with different edge labels due to an overused number of codes. And a potential consequence could be that you need more data to get sufficient frequent subgraph coverage.
>
> Overall, we agree that broader threshold–complexity trade-off curves would be valuable, but we view the current controlled sweep as the right first step for isolating the sparsity question under a fixed discrete representation.
>
> > Q: The unconditional generation results show a strong bias toward alpha-helical structures (~90% helix content). What steps could mitigate this for more diverse topologies? Fuller exploration on what's the root cause of the strong preference would be interesting.
>
> Please refer to our general response "Concern on strong alpha-helix overrepresentation".
>
> [1]: Rives, Alexander, et al. Biological structure and function emerge from scaling unsupervised learning to 250 million protein sequences. PNAS 2021.
>
> [2]: Bissantz, Caterina, Bernd Kuhn, and Martin Stahl. A medicinal chemist’s guide to molecular interactions. Journal of Medicinal Chemistry 2010.

---

### Author Response · Authors · 2025-11-24
**General response**

We thank all reviewers for their thoughtful feedback, which has strengthened our paper.

All changes we made to the manuscript are highlighted in blue. We first address some common points raised by multiple reviewers.

> Concern about strong alpha-helix overrepresentation

Multiple reviewers noted the strong alpha-helix bias in our generated samples, which we also acknowledged in the original submission. After careful analysis, we believe this effect stems from two factors:

- **Inference hyperparameters.** Our initial inference settings (top-p and temperature) were chosen to maximize co-designability. This objective can skew the sampled distribution toward highly designable local patterns, and in practice leads to a more helical secondary-structure profile.
- **Training data.** We train on the same AFDB subset used by Genie2 and Proteina. Training on this dataset generally leads to models with strongly helix-dominant samples (see Table 1 in [2]). In particular, La-Proteina trained on the same data also shows similarly low beta-sheet proportions (~3–4%; see Appendix C.1 in [1]). This suggests the bias is largely caused by the data rather than being unique to AutoFold.

To quantify and partially mitigate both effects, we performed two controlled follow-up experiments:

- **Inference re-tuning on the original dataset.** We re-selected inference hyperparameters with the explicit goal of increasing beta-sheet content.
- **Training on La-Proteina's dataset.** We retrained AutoFold-m on the secondary-structure-filtered AFDB subset used by La-Proteina [1] (denoted AFDB-LaProteina), yielding AutoFold-m*. Note that at the time of submission their dataset was not yet released.

The resulting metrics are summarized below.

| Method | Co-designability (%) ↑ | | Diversity (normalized) ↑ | | | Novelty ↓ | | Designability (%) ↑ | Sec. Struct. (%) | |
|--------|---------|-----|-----|-----|----------|-----|------|------------|----|----|
| | Unique | all | Str | Seq | Seq+Str | PDB | AFDB | PMPNN-8 | α | β |
| La-Proteina-1 | 41.2 | **72.2** | 0.571 | 0.598 | 0.834 | 0.75 | 0.82 | 93.8 | 72 | 5 |
| La-Proteina-2 | 36.0 | 59.6 | 0.604 | 0.634 | 0.836 | 0.77 | 0.86 | **94.6** | 63 | 10 |
| **Tuned to optimize co-designability** | | | | | | | | | | |
| AutoFold-t | 36.2 | 50.4 | **0.718** | **0.984** | _0.952_ | 0.16 | 0.19 | 58.0 | 89 | 2 |
| AutoFold-s | _42.4_ | 68.8 | 0.616 | 0.910 | 0.945 | **0.10** | **0.12** | 69.4 | 91 | 2 |
| AutoFold-m | **46.4** | _72.0_ | 0.644 | _0.950_ | **0.986** | _0.14_ | _0.17_ | _69.6_ | 90 | 1 |
| **Tuned to maximize beta-sheet content** | | | | | | | | | | |
| AutoFold-t | 29.2 | 45.4 | 0.643 | 0.982 | 0.956 | 0.16 | 0.19 | 52.2 | 89 | 2 |
| AutoFold-s | 37.6 | 55.0 | 0.618 | 0.938 | 0.978 | 0.19 | 0.23 | 57.0 | 88 | 2 |
| AutoFold-m | 39.2 | 58.2 | 0.674 | 0.959 | 0.966 | 0.24 | 0.26 | 60.6 | 85 | 4 |
| AutoFold-m* | 32.0 | 48.0 | 0.667 | 0.938 | 0.967 | 0.48 | 0.52 | 46.8 | 69 | 12 |

Here, AutoFold-m* is trained on AFDB-LaProteina. La-Proteina-1/2 correspond to the same model with two different hyperparameter settings, where La-Proteina-2 achieves higher beta-sheet content. Additionally, we provide visualization of the generated samples in Appendix F.1, and the secondary structure profile plots (similar to those conducted by FoldFlow-2) in Appendix E.3 in the revised manuscript.

Our new results suggest that:

- **Decoding matters, but the dataset is the dominant driver.** When we retune decoding to favor beta-sheets on the original dataset, AutoFold reaches beta-sheet proportions comparable to prior work (≈3–4%) while maintaining strong unique co-designability, still outperforming baselines in our Table 1, though slightly below the strongest La-Proteina setting.
- **Training on AFDB-LaProteina substantially improves the -alpha-helix/beta-sheet balance.** AutoFold-m* attains a secondary-structure profile similar to La-Proteina-2, indicating that much of the helical bias is not intrinsic to our AR contact-graph formulation.
- **Diversity remains a consistent strength.** Across all settings, AutoFold retains substantially better diversity, especially in sequence diversity and sequence FID (Sec. 5.2). This suggests that our discrete representation captures the sequence distribution more faithfully.
- **Novelty stays high even when the SSE balance is corrected.** While extreme helix-rich samples can inflate novelty by producing long helices, AutoFold-m*, with a helix/beta balance close to La-Proteina, still achieves strong novelty, indicating that high novelty is not solely an artifact of long helical overproduction.

We have incorporated this analysis and new results into the revision (Section 5.2).

Finally, please note that due to the limited time in this cycle, we could not conduct the same experiments on AFDB-LaProteina for AutoFold-s and AutoFold-t, but we believe those experiments would not affect our conclusion and we will update our results once we finish all model evaluation.

---

> ### Author Response · Authors · 2025-11-24
> **General response (cont.)**
>
> > Concern on potentially unfair comparison due to length-unconstrained generation
>
> We agree that length-unconstrained sampling can bias unconditional metrics and complicate direct comparisons to diffusion/flow matching baselines that are length-conditioned. However, we believe our benchmark provides a useful middle ground to compare our model, **the first AR model for protein sequence-structure co-generation**, to existing diffusion/flow matching models.
>
> To give an idea of the length distribution, we include length histograms of co-designable samples from all AutoFold variants, alongside the corresponding histograms for the training sets. These plots show that AutoFold largely matches the training length distribution, with a modest shift toward shorter proteins. This implies that our current models may be less effective than La-Proteina at generating long co-designable proteins, and we acknowledge this limitation transparently. At the same time, we view it as a practical rather than fundamental constraint of the approach, tied to training scale and engineering choices.
>
> We also want to clarify the intended scope and positioning of this work, and we have revised the manuscript accordingly. **To better reflect that our contribution is not primarily about speedups or long-protein unconditional generation, we have removed “ultra-fast” from the title and revised the introduction to emphasize the methodological focus**. In particular, our key contributions are:
>
> - **First proof-of-concept for autoregressive contact-graph protein generation.** Our goal is not to claim unconditional state-of-the-art performance at this stage, but to demonstrate that proteins can be generated efficiently via sparse contact-graph modeling in an autoregressive framework. The strong diversity and novelty we observe, together with this discrete graph perspective, open a new modeling avenue for protein generation.
> - **A single model for both unconditional and substructure-conditional generation.** AutoFold trains one autoregressive model that supports unconditional sampling and motif scaffolding through the same generative mechanism, without task-specific retraining, while maintaining a reasonable designability regime.
> - **Promising performance on a real-world downstream task.** On motif scaffolding, a more practically relevant setting than unconditional generation, we achieve performance comparable to La-Proteina, despite using substantially fewer computational resources. This highlights the value of our formulation beyond unconditional benchmarks.
>
> Overall, we acknowledge the reviewers' point and believe the added length analyses make the comparison clearer. We hope this positioning also clarifies that our contribution is to establish a viable contact-graph generation paradigm, and to show its competitive behavior on conditional design tasks, rather than to optimize all unconditional metrics in this first study.
>
> [1]: Geffner, Tomas, et al. La-proteina: Atomistic protein generation via partially latent flow matching. arXiv preprint 2025.
>
> [2]: Geffner, Tomas, et al. Proteina: Scaling Flow-based Protein Structure Generative Models. ICLR 2025.

---

> > ### Author Response · Authors · 2025-12-01
> > **New revision summary**
> >
> > We thank all reviewers again for their time and thoughtful feedback. We have uploaded a new revision that incorporates the following changes:
> >
> > - We added the results for DPLM-2 and MultiFlow to Table 1, as requested by reviewers dGv6 and Cd7Q. Our claims on AutoFold's performance remain.
> > - We added the results for AutoFold-t* and AutoFold-s*, i.e., AutoFold models trained on the AFDB-LaProteina dataset. These results further affirm that the training dataset is the main driver of the alpha-helix overrepresentation.

---

### Meta-Review · Area_Chair_LsML · 2026-01-08

**Summary:**

This paper introduces AutoFold, a two-stage generative framework that discretizes protein backbones into sparse contact graphs via a VQ-VAE and subsequently generates them using an autoregressive model following AutoGraph. The initial reviews were divided, with one enthusiastic accept (8, but review is a bit short), one weak accept (6), and three borderline ratings (4) that raised significant methodological and experimental concerns.

While reviewers recognize the novelty of shifting from continuous coordinate diffusion to discrete AR graph generation and the resulting inference speed, major concerns focused on: (1) a severe bias toward alpha-helices (>90%) and lack of beta-sheets, (2) unfair comparisons against baselines due to unconstrained generation lengths (AutoFold generates shorter proteins) (3) the limitation of modeling only the backbone rather than all-atom structures and unfair compairson to all-atom motif scaffolding, and (4) exaggerated claims regarding speedups ("order of magnitude") (5) lack of experimental details and unmatched training dataset.
The reviewers also note the model's better novelty is a tradeoff for lower overall designability, which makes the advantage compared to La-Proteina unclear.

**Reviewer Concerns:**

The rebuttal addressed some of the major concerns regarding alpha-helix and clarity, but some concerns are not fully addressed, especially the designability metrics are confounded by the model's bias towards shorter proteins and extremely high helix percentages.

**Addressed Concerns**
- Alpha-Helix Bias (jUAk, dGv6, Cd7Q, pLGR): The authors provided concrete evidence that this bias stemmed from the training set (AFDB). They retrained a variant (AutoFold-m*) on the balanced "La-Proteina" dataset, improving beta-sheet content from ~1% to 12%, matching the distribution of La-Proteina. However, under the matched setup, the co-designability is lower than La-Proteina (authors argue that novelty is better).
- Methodological Clarity (dGv6, pLGR): The authors revised the text to explain the graph flattening process (SENT), tokenization scheme, and stochastic ordering, and provided more details about the dataset.
- Speedup Claims (dGv6, NzJQ): The authors retracted the "order of magnitude" claim relative to flow-matching models (La-Proteina), clarifying the speedup is ~3x, and removed "Ultra-fast" from the paper title.
- Missing Baselines (Cd7Q, dGv6): The authors added MultiFlow and DPLM-2 to the tables.

**Partially Addressed**:
- Unfair Length Comparisons (dGv6, Cd7Q, NzJQ): The authors provided histograms confirming their model indeed generates shorter proteins than the baselines, acknowledging the advantage this gives in designability metrics. However, they did not re-run the primary benchmarks with length-matched constraints, leaving the direct comparison in Table 1 slightly confounded.

**Outstanding**:
- Backbone vs. All-Atom (NzJQ, Cd7Q): The reviewers criticized the model for being backbone-only while SOTA (La-Proteina) is all-atom. The authors maintained that this is a scope limitation and a valid first step for the new architecture.
- Designability vs. Novelty trade-off (NzJQ): There remains a philosophical disagreement on whether high novelty scores justify significantly lower designability (foldability) scores compared to La-Proteina.

**Reviewer Scores:**

- Reviewer jUAk (Original: 8): Predicted: 8 (Accept). The primary concern (helix bias) was methodically resolved with the retraining experiment.
- Reviewer dGv6 (Original: 4): Actual: 6 (Marginally Above). The reviewer explicitly stated in the discussion that the clarifications on the method and helix bias warranted a score increase.
- Reviewer Cd7Q (Original: 4): Predicted: 4 (Marginally Below). While the helix bias was fixed, the "unfair" length comparison remains.
- Reviewer NzJQ (Original: 4): Predicted: 4 (Marginally Below). This reviewer is skeptical of the result due to novelty-designability tradeoff. The rebuttal addressed the helix bias, but did not show the requested result for protein with >300 residues.
- Reviewer pLGR (Original: 6): Predicted: 6 (Accept). The rebuttal clarified the "why faster inference?" question and the hyperparameter choices, but there is likely no significant new results that could lead to a further increase.

---

### Decision · Program_Chairs · 2026-01-26

Reject